

# A Global land snow scheme (GLASS) v1.0 for the GFDL Earth System Model: Formulation and evaluation at instrumented sites

Enrico Zorzetto[1,2], Sergey Malyshev[3], Paul Ginoux[3], and Elena Shevliakova[3]

[1]Program in Atmospheric and Oceanic Sciences, Princeton University, Princeton, NJ, USA
[2]Earth and Environmental Sciences Department, New Mexico Institute of Mining and Technology, Socorro, NM, USA
[3]NOAA OAR Geophysical Fluid Dynamics Laboratory, Princeton, NJ, USA

**Correspondence:** Enrico Zorzetto (ez6263@princeton.edu)

**Abstract.**

Snowpack modulates water storage over extended land regions, and at the same time plays a central role in the surface albedo feedback, impacting the climate system energy balance. Despite the complexity of snow processes and their importance for both land hydrology and global climate, several state-of-the-art land surface models and Earth System Models still employ relatively simple descriptions of snowpack dynamics. In this study we present a newly-developed snow scheme tailored to the Geophysical Fluid Dynamics Laboratory (GFDL) Land Model version 4.1. This new snowpack model, named GLASS ("Global LAnd-Snow Scheme"), includes a refined and dynamical vertical layering snow structure which allows us to track in each snow layer the temporal evolution of snow grain properties, while at the same time limiting the model computational expense, as necessary for a model suited to global-scale climate simulations. In GLASS, the evolution of snow grain size and shape is explicitly resolved, with implications for predicted bulk snow properties, as they directly impact snow depth, snow thermal conductivity and optical properties. Here we describe the physical processes in GLASS and their implementation, as well as the interactions with other surface processes and the land-atmosphere coupling in the GFDL Earth System Model. The performance of GLASS is tested over 10 experimental sites, where in-situ observations allow for a comprehensive model evaluation. We find that, when compared to previous version of GFDL snow model, GLASS improves predictions of seasonal snow water equivalent and soil temperature under the snowpack.

## 1 Introduction

Snow is a fundamental component of the global water and energy balance. Over extended regions on Earth, a significant fraction of the water budget is stored over land as snow, so that soil moisture, runoff and water availability for ecosystems and human communities are directly impacted by changes in snowpack (Cohen and Rind, 1991; Xu and Dirmeyer, 2013). Snow cover also plays an important role in the energy balance at the surface (Qu and Hall, 2014; Thackeray et al., 2018). Compared to other natural surfaces, snow is characterized by the highest reflectivity in the visible range, and by an exceptionally low heat





conductivity. Because of these properties, snow has been shown to significantly affect near surface temperatures (Armstrong and Brun, 2008; Betts et al., 2014) and to play a primary role in modulating the warming rate of arctic regions (Stieglitz

et al., 2003) and permafrost extent (Burke et al., 2013). Thus, the presence of snow fundamentally alters the near-surface temperature and in turn the energy partitioning between land surface, subsurface and atmosphere (Henderson et al., 2018). Numerical simulations of snowpack are used in many scientific applications, ranging from and watershed-scale hydrology and flood forecasting (Nester et al., 2012; Blöschl, 1999) to centuries-long, global simulations of the climate system (Kapnick and Delworth, 2013). Given the profound implications of snow for land-atmosphere interactions over extended regions of Earth, it

is paramount that land surface models adequately describe the coupling between snow and soil, vegetation, and the atmosphere.

Fully understanding the implications of snow for land hydrology and the climate system requires a detailed representation of its physical properties in numerical models. The complexity of snow schemes used in land surface modelling varies greatly and has been previously classified in three complexity levels (Boone and Etchevers, 2001; Vionnet et al., 2012). The first class includes the simplest snow models, which either consist of a single snow layer, or of a composite snow-soil medium.

Traditionally these computationally inexpensive snow models have been employed in numerical weather prediction and global climate models. The second class of "intermediate complexity models" addresses several deficiencies of the former class of models by including at least a coarse vertical discretization of the snowpack, and by explicitly modelling the liquid phase water and variations in snow density. Intermediate-detail snowpack models include the ECMWF snow scheme (Dutra et al., 2010; Arduini et al., 2019), the Community Land Model (CLM) 4.5 (Oleson et al., 2013), the Canadian Land Surface Scheme

(CLASS), JULES (Best et al., 2011), Snow17 (Anderson, 1976), and WEB-DHM-S (Shrestha et al., 2010).

Finally, the third class consists of "detailed snowpack models": These are characterized by a much finer vertical layering of the snow, which can evolve dynamically with snowfall and snow melt. Snow microphysical properties are tracked in each snow layer, thus allowing for a more realistic description of physical processes. Such highly-detailed snowpack models include SNOWPACK (Bartelt and Lehning, 2002; Lehning et al., 2002b, a), SNTHERM89 (Jordan, 1991), and CROCUS (Brun et al.,

1992, 1997; Vionnet et al., 2012). Some models explicitly resolve the propagation of shortwave radiation within the snow layers (For example, SNICAR (Flanner and Zender, 2005; Flanner et al., 2007), TARTES (Libois et al., 2013) and GEMB (Gardner et al., 2023)). Since higher-detail snow schemes tend to be computationally expensive, applications of snow models targeting long, global-scale numerical simulations of the Earth system must strike a balance between physical detail and computational demands. Despite the need for this trade-off, it has been recognized that a number of physical processes impacting the evolution

of snowpack should be resolved in land surface models, as they can be relevant for large–scale hydrological studies and for coupled climate simulations.

The increasing fidelity of snow processes using detailed snow schemes has been shown to benefit both climate studies (Dutra et al., 2010; Decharme et al., 2016) and numerical weather predictions applications (Arduini et al., 2019), as it does not only impact the hydrological response, but also interaction with the atmosphere through surface temperature and reflectivity.

The snow scheme currently implemented in the GFDL LM 4.1 can be considered a scheme of intermediate complexity: The snowpack is characterized by a fixed number of vertical layers (routinely set to 5) each characterized by temperature, ice and liquid water content. However, the density of the snow is set to a constant value (currently, $250\,\mathrm{kg\,m^{-3}}$), so that the model



provides limited information on snow depth. As a consequence, snow heat conductance is also a constant, which can lead to challenges in determining the vertical temperature profile of snow and soil. Finally, no description of snow microphysics is present, so that dependence of physical processes on the snow micro structure (e.g., the evolution of snow optical properties with age and snow compaction) is not accounted for. However, this parsimonious snowpack model has been successfully employed to simulate snow cover at the global scale (Kapnick and Delworth, 2013).

The focus of this work is to present the Global LAnd Snow Scheme (GLASS), a novel snow model developed for LM 4.1. The primary objective of the development of GLASS is to increase the realism of the snow processes, while at the same time limiting the computational burden of the model so that it can be effectively employed in global Earth system simulations. Key physical processes that were absent in LM4.1 have been adapted in GLASS from existing parameterizations used in detailed snow schemes. In particular, GLASS now includes the treatment of snow compaction, wind drift effect, and snow aging, and accounts for the effects of these processes on snow thermal and optical properties. The evolution of snow properties with snow aging accounts for both dry and wet metamorphism. In GLASS these processes affect not only the growth of snow grains, but also the evolution of their shape. This information is in turn employed for evaluating snow albedo, which in GLASS depends explicitly on both optical size and optical shape of snow grains. While increasing the fidelity of the snow physical processes, GLASS builds on the existing implicit solution scheme for the fluxes between land and atmosphere, which is numerically stable and efficient for the time step (30 minutes) routinely used in global-scale coupled land-atmosphere simulations. To avoid an excessive increase in the computational demands of the new snow model, the energy balance at the land surface is linearized in LM 4.1. This approach leads to a trade-off between computational cost and physical realism, as an iterative solution of the energy balance would lead to a considerable increase in computational expense. GLASS is characterized by a dynamic snow vertical layering structure which allows to efficiently track the evolution of snow properties with age, as is currently done by a few high detail snow schemes such as CROCUS (Brun et al., 1992; Vionnet et al., 2012). A relayering scheme is used to determine the optimal vertical discretization of the snowpack, so as to obtain a proper trade-off between model detail and computational expense.

After presenting the features of GLASS we test its performance over a set of sites widely used as benchmark, including in snow model intercomparison efforts (Krinner et al., 2018). The data set used here spans a wide range of climate and terrain conditions, so as to characterize the behavior of the model. The evolution of snowpack in vegetated areas constitutes a major source of uncertainty, which carries potentially large implications for constraining the snow albedo feedback over the Northern Hemisphere. We contribute to this challenge by evaluating GLASS at three forested sites that are part of the ESM-SnowMIP project.

The remainder of the manuscript is organized as follows: Section 2 describes in the detail the model physics and implementation in LM 4.1, including the existing treatment of snow processes. The GLASS is presented in detail in Section 3. Section 4 describes the experimental setup used in our study as well as the data used as model input, atmospheric forcing and snow data used for model validation. Results and discussion follow in Sections 5 and 6, while model limitations and considerations for future research directions are discussed in Section 6. Conclusions from this study are featured in Section 7.





## 2 Overview of land and snow processes represented in the GFDL Land Model

### 2.1 Land model overview

The land component of the GFDL ESM 4.1 (Dunne et al., 2020), hereafter termed LM4.1, provides a detailed description of
the key processes involved in the mass and energy exchanges between land and atmosphere.

In LM4.1, the land domain is discretized in a number of grid cells. To represent the effects of heterogeneity of land-atmosphere interactions and terrestrial biogeochemical processes, the model employs a mosaic approach where each grid cell can be further split into a set of sub-grid tiles: fractions of the grid cell with distinct physical and biogeochemical properties. LM4.1 resolves the land-atmosphere exchanges of energy, water, and tracers separately for each of the tiles. The evolution of
the relevant terrestrial properties — such as the state of vegetation, albedo, soil moisture and temperature, snow cover, etc. — is also simulated separately for each of the tiles, while allowing for interaction due to land use transitions and other processes that can dynamically change tiling structure. Such an approach captures the effects that land use has on land-atmosphere physical interactions, as well as on the terrestrial carbon cycle (Shevliakova et al., 2009; Malyshev et al., 2015; Chaney et al., 2018; Zorzetto et al., 2023).

Vegetation in the model is dynamic, represented by a set of cohorts, with each cohort being a set of plants with similar characteristics, i.e. species, size, and age. Cohorts change as the vegetation assimilates carbon (and undergoes other processes, such as mortality, reproduction, etc.), and organize themselves in a number of layers, according to the Perfect Plasticity Approximation (PPA) approach (Strigul et al., 2008; Weng et al., 2015; Martínez Cano et al., 2020).

In the present application, we employ a configuration of the model, where we focus on a single tile that corresponds to each
of the sites where point-scale observations were obtained. The time step used in the model for physical processes related to snow, soil and land-atmosphere interactions is 30 minutes.

In the GFDL LM 4.1, sensible ($H_g$) and latent ($E_g$) heat fluxes are computed using the bulk formulas driven by the gradient in temperature and specific humidity between atmosphere ($T_a$, $q_a$) and the near-surface "*canopy air*" layer ($T_c$,$q_c$)

$$H_g = \rho_{air} c_p C_D \overline{U} \left( T_c - T_a \right) \tag{1}$$


$$E_g = \rho_{air} C_D \overline{U} \left( q_c - q_a \right) \tag{2}$$

where $\overline{U}$ is the wind above the constant flux layer and $C_D$ is the stability-dependent drag coefficient computed from the Monin-Obukhov similarity theory (Garratt, 1994; Foken and Napo, 2008), $c_p$ is the specific heat of air, and $\rho_{air}$ is the air density. Within the vegetation canopy, the aerodynamic resistance is computed assuming an exponential wind velocity profile
following the approach by Bonan (1996). In the present work we use the same formulation for turbulent fluxes in all model configurations described below.





## 2.2 Snow scheme in LM4.1

The existing snow module part of GFDL LM4.1 (termed "Current Model", LM-CM or CM in throughout this work) can be classified as an intermediate complexity snow scheme according to the definition of Boone and Etchevers (2001). Fluxes of water and heat in the snowpack and soil continuum are based on the model by Milly et al. (2014). If snow is present on the ground, the snowpack is composed of a fixed number of levels, routinely set to 5. Each snow layer is characterized by snow temperature, and both liquid and ice mass content. No description of snow microphysics is present, so that key snow properties (in particular, snow density and heat conductance) are assumed to be constant. Light does not penetrate the snowpack: Short-wave radiation contributes to the surface energy balance, and the resulting net heat flux constitutes the boundary condition for resolving the heat diffusion through the snow layers and the underlying soil. The snow albedo is computed with an empirical formulation based on the Bidirectional Reflectance Distribution Function (BRDF) described below in Appendix C. This model does not explicitly account for the effects of snow grain size or for the presence of light absorbing particles within the snow, but rather yields typical reflectivity values for snow-covered surfaces. The effect of snow metamorphism on its optical properties is mimicked by introducing a dependence of the BRDF model parameters on the temperature of the snow uppermost layer, with warm snow being characterized by reduced reflectivity.

## 3 The Global Land-Atmosphere-Snow Scheme (GLASS) in LM4.1

GLASS was developed building upon the existing GFDL snow scheme with the objective of including representations of important snow physical processes into the model. For each LM4.1 model tile, GLASS is a 1D snow model coupled to the soil and multi-layer canopy schemes included in LM. GLASS simulates the evolution of snowpack and the exchanges of water and energy with the lower atmosphere and the underlying soil. The vertical discretization of the snowpack is dynamic, with the number and thickness of snow layers being determined both by history (e.g., new snow layers being added on top of the snowpack due to precipitation events) and by computational considerations, so that snow layers can split and merge tending to an optimal vertical profile. In GLASS, each snow horizontal layer is characterized by a number of physical properties. A schematic representation of the snow vertical structure in GLASS is shown in Figure 1. Energy and mass fluxes at the upper boundary of this medium are determined by liquid and solid precipitation (possibly percolating through canopy layers, if vegetation is present), evaporation or sublimation, and the net heat flux into or out of the snowpack is determined by solving the energy balance at the surface. Shortwave radiation can penetrate the snowpack depending on its thickness and optical properties, as discussed below. At the bottom of the snow column, the boundary condition is given by the flux of heat and water into the underlying soil layers, or by runoff.

### 3.1 Representation of snow at the ground and its vertical discretization

Design of snow layers is essential to adequately capture the vertical variation of key snow physical properties which affects the overall snowpack mass and energy balance. In GLASS, this requirement is met by employing a vertical structure which can





| Name | Variable | Units | Prognostic | Size |
|---|---|---|---|---|
| $\Delta z_k$ | Thickness | m | Yes | 1 |
| $w_{l,k}$ | Liquid content | $\mathrm{kg\,m^{-2}}$ | Yes | 1 |
| $w_{s,k}$ | Ice content | $\mathrm{kg\,m^{-2}}$ | Yes | 1 |
| $T_k$ | Temperature | K | Yes | 1 |
| $d_{opt,k}$ | Optical diameter | m | Yes | 1 |
| $\delta_k$ | Snow dendriticy | dimensionless | Yes | 1 |
| $s_{p,k}$ | Snow sphericity | dimensionless | Yes | 1 |
| $H_{sn,k}$ | Heat content | $\mathrm{J\,m^{-2}}$ | Derived | 1 |
| $\lambda_k$ | Heat conductance | $\mathrm{J\,m^{-2}\,K^{-1}}$ | Derived | 1 |
| $c_k$ | Heat capacity | $\mathrm{W\,m^{-1}\,K^{-1}}$ | Derived | 1 |
| $\rho_k$ | Snow density | $\mathrm{kg\,m^{-3}}$ | Derived | 1 |
| $age_k$ | Snow age | days | Yes | 1 |

**Table 1.** List of physical variables characterizing the $k^{\mathrm{th}}$ snow layer.

change dynamically, designed to strike a trade-off between the desire of physical detail, and the need to limit the computational requirements of a model used for global-scale simulations. The snowpack, if present, is composed of a variable number $N_L$ of horizontal layers, numbered from the top of the snowpack to the bottom ($k = 1, \dots, N_L$). Each snow layer is characterized by a set of physical properties which evolve dynamically. These are the layer's liquid ($w_{l,k}$) and ice ($w_{l,k}$) contents, its thickness $\Delta z_k$, temperature $T_k$, heat capacity $c_k$, and heat conductance $\lambda_k$. In each layer, we assume that ice and liquid water components of the snowpack have the same bulk temperature, which can thus be determined by a single heat conservation equation. Additionally, the physical properties of snow grains in each layer are described by three prognostic variables: The snow grain dendriticy $\delta_k$, sphericity $s_{p,k}$, and optical diameter $d_{opt,k}$. Together, these three prognostic variables identify optical size and shape of snow grains and are used for albedo calculations. A complete description of the physical properties of each snow layer is provided in Table 1.

In GLASS, the thickness of the snow layers is adaptive to the snow depth and the thermal regime within the snowpack. To accurately represent the snow thermal conductivity, the layers are generally thinner in the region of the high thermal variability of gradients (e.g. near the surface, which is subject to high-frequency variation of fluxes, or in the vicinity of soil surface) and thicker in the middle of the snowpack. The number and thickness of layers is dynamic. Snowfall events of large enough magnitude can lead to the creation of new layers on top of existing snow or bare soil, according to the rule:

$$\Delta n_L = \max\left[\max\left(3 - n_L, 1\right), \min(5, \lceil a_{fall} \cdot \Delta z_{fall}\rceil)\right] \tag{3}$$

where $\Delta z_{fall}$ is the depth of newly fallen snow ([m]) and $a_{fall} = 100\,\mathrm{m^{-1}}$.

Eq. (3) ensures that after snowfall the number of snow layers $n_L + \Delta n_L$ is at least 3. The number of newly created layers $\Delta n_L$ originated from the snow falling in a single time step can be up to 5, depending on the magnitude of the precipitation event.



In case of weak precipitation event, instead of creating new layers according to Eq. (3), the precipitating snow mass is added to the existing snowpack uppermost layer, if any (this happens if the newly deposited snow depth is up to half the depth of the uppermost existing snow layer). In addition to snowfall, other physical processes (e.g., sublimation, snow compaction, snow melt) can modify the thickness of existing snow layers. To avoid dealing with a snowpack composed of an excessive number of thin layers, the model performs at each time step re-layering of the snowpack, with the objective of avoiding excessive costs in compute time and memory as well as potential numerical instabilities originating from dealing with very small snow layers resulting from snow melt or sublimation.

The model optimizes the vertical layering of the snowpack by first defining an optimal distribution of snow layers: the optimal thicknesses of the top and the bottom layers (nearest to the snow/soil interface) are specified as model input. Below the first specified top layer, the optimal layers increase in thickness with a given constant ratio (set to 1.5 in the current model configuration), until they reach a specified maximum thickness (1 m in the default configuration used here). The actual snowpack vertical structure is then compared with the optimal one and differences are minimized through merging and splitting of existing snow layers. This procedure is performed until the entire depth of the snowpack is covered, and then the vertical distribution of layers is scaled to ensure an integer number of layers in snowpack. There is no maximum number of snow layers set in the model. Two layers can be merged only if their physical properties are not too dissimilar. In the current application, we allow the merging of layers $k$ and $k+1$ only if their differences in grain sphericity, optical diameter and density are below given thresholds (namely, $|s_{p,k} - s_{p,k+1}| < 0.2$, $|d_{opt,k} - d_{opt,k+1}| < 1 \times 10^{-4}$ m, and $|\rho_k - \rho_{k+1}| < 30$ kg m$^{-3}$). Thus, in the process of merging snow layers for computational purposes, the vertical heterogeneity of snow physical properties is taken into account and preserved to some extent.

To split the layer $k$, GLASS examines a penalty function $\mathcal{P}$ that, given boundaries of the layers $z_k, k = 0 \ldots N_L$, returns a value indicating how far the current distribution is from the optimal configuration:

$$\mathcal{P} = \sum_{i=0}^{N-1} \left( z_{i+1} - z_i - \mathcal{D}(z_i) \right)^2 \tag{4}$$

where $\mathcal{D}(z)$ is the optimal layer thickness at depth $z$. If $\mathcal{P}(\ldots, z_k, z_k + \mathcal{D}(z_k), \ldots) < \mathcal{P}(\ldots, z_k, \ldots)$, then layer $(z_k, z_{k+1})$ is split in two, $(z_k, z_k + \mathcal{D}(z_k))$ and $(z_k + \mathcal{D}(z_k), z_{k+1})$. Similarly, if $\mathcal{P}(\ldots, z_k, z_{k+2}, \ldots) < \mathcal{P}(\ldots, z_k, z_{k+1}, z_{k+2}, \ldots)$ and if the layers are not otherwise prohibited from merging (as it can be required if they have significantly different physical properties) then the layers $(z_k, z_{k+1})$ and $(z_{k+1}, z_{k+2})$ are merged into one. To obtain the optimal thickness $\mathcal{D}(z)$ at any given depth $z$ given the discrete values of optimal layer thicknesses described above, the model builds piecewise linear dependence of "layer number" $\mathcal{L}(z)$ and its inverse dependence of $z$ on "layer number" $\mathcal{Z}(L)$. Then the optimal thickness for layer at depth $z$ is the distance from current point to the point where layer number is greater than current by exactly 1, such that $\mathcal{D}(z) = \mathcal{Z}(\mathcal{L}(z) + 1) - z$.

At each time step, the vertical layering profile of the snow is compared to this theoretical optimal profile. The distance of each layer from the optical thickness at that depth is compared to that of the profiles obtained splitting and merging each layers with its neighbours. If such operations lead to a vertical profile which is closer to the optimal one, the split or merge operation



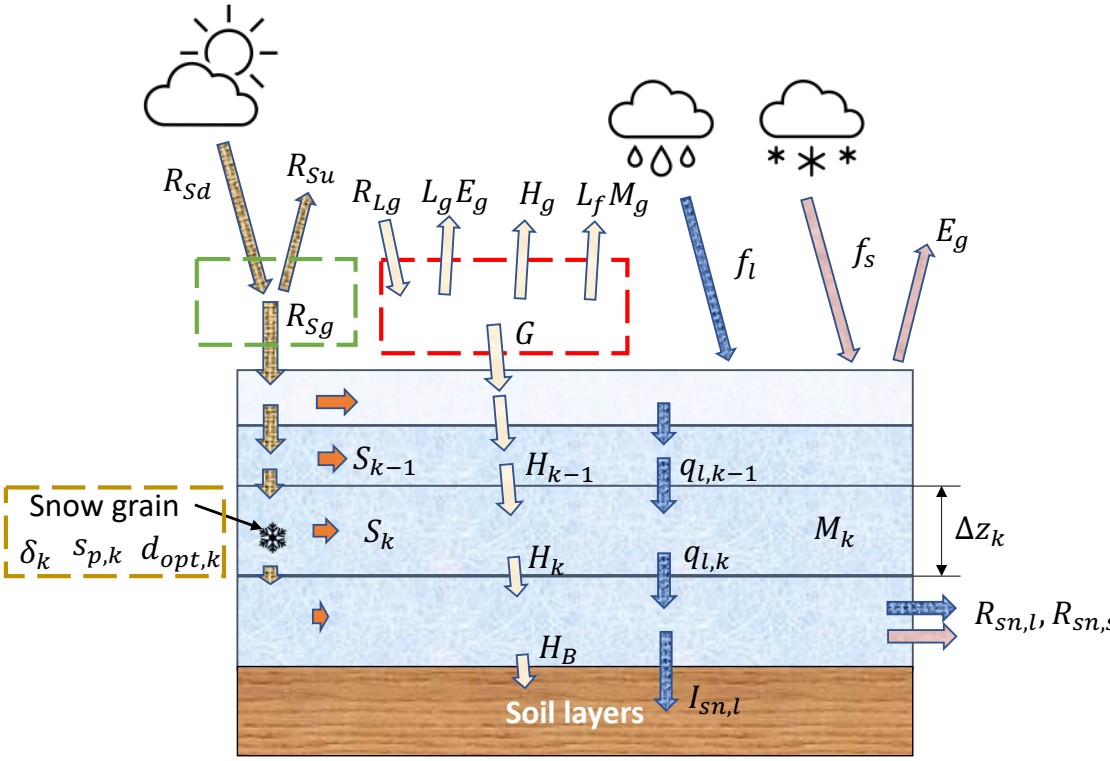

**Figure 1.** A schematic representation of the fluxes of energy and water included in the GLASS snow model. The red and green dashed rectangles indicate the energy flux terms contributing to the surface energy balance, and the shortwave radiative balance at the surface, respectively. Exchange terms include net longwave ($R_{Lg}$) and net shortwave ($R_{Sg}$) radiation (with downward and upward radiation denoted by $R_{Sd}$ and $R_{Su}$, respectively), as well as sensible heat flux ($H_g$) and latent heat of evaporation ($L_g E_g$) and due to the melting rate at the ground ($M_g$). Solid ($f_s$) and liquid ($f_l$) precipitation rates are also represented. Heat fluxes between layers ($H_k$) and heat sources due to shortwave radiation absorption ($S_k$) within the snowpack are also represented, as are liquid ($R_{sn,l}$) and solid contributions to runoff ($R_{sn,s}$) and liquid water infiltration into soil ($I_{sn,l}$). Fluxes of liquid water (blue arrows) and ice (pink arrows) are also featured, with $q_k$ for liquid flow between the snowpack layers $k$ and $k+1$. Note that fluxes of heat advected by solid and liquid precipitation, vertical liquid water flow within the snowpack, infiltration and runoff are all represented in the model although not explicitly represented in this figure for simplicity.



is performed on the snow layers. While there is no upper limit on the number of layers, this approach allows us to control the
evolution of the snow vertical profile.

## 3.2 Energy balance at the surface

Figure 1 provides a schematic representation of the physical processes included in the snow model. The energy balance at
the surface is coupled with that of the snow or soil underneath, and the vegetation layers and canopy air above, and can be
expressed in units of $[\mathrm{W\,m^{-2}}]$ as

$$R_{sg} + R_{lg} - H_g - L_g E_g - G - L_f M_g = 0 \tag{5}$$

where $R_{sg}$ and $R_{lg}$ are the shortwave and longwave net radiative fluxes at the ground surface, $L_f$ and $L_g$ are the latent heats of
fusion and evaporation, respectively, $E_g$ the rate or evaporation or sublimation, $G$ heat flux in or out of the ground, and $M_g$ the
melting rate of water at the surface (if $> 0$, else it is the freezing rate). In the presence of deep enough snowpack ( $> 0.05\,\mathrm{m}$
by default), the net shortwave radiation is absorbed within the snowpack instead of contributing to the surface energy balance.
Similarly, the latent heat carried by precipitation is accounted for in the energy balance of the underlying snowpack or soil
where precipitation accumulates. Equation (5) is solved together with the equations of mass and energy balance of canopy air,
the energy balance of any vegetation canopy layers, and the mass balance of any liquid or solid water intercepted by canopy
layers.

In order to run efficiently in long global-scale simulations, the solution of this system of equations must avoid excessive
computational costs and be numerically stable for relatively large time steps (the present application uses a 30-minute time step,
which corresponds to physics time step in typical GFDL atmospheric model configuration). To avoid numerical instabilities,
the system is solved using a fully implicit scheme. This is done by linearizing the system of equations around the current
value of its prognostic variables. These are the temperatures of ground, vegetation canopies and canopy air, water/snow mass
intercepted by the canopies, and the specific humidity of canopy air. The solution of this system of equations conserves energy
and water mass as required by long Earth system simulations. However, the rate $M_g$ of water melt or freeze at the surface of the
snowpack (if present) or at the ground surface (if snow is absent) imposes a significant non-linearity in eq. (5), since this term
is constrained by the amount of liquid or frozen water which can undergo phase change in the snowpack or in the upper soil
layer. In this case the single nonlinear eq. (5) is solved in order to obtain the change in temperature at the surface of the ground
(or, of the snowpack if present, the temperature at the top of the snowpack), $T_g$, which in turn is used to obtain the tendencies
of all other prognostic variables of the problem from the linearized system. The solution of eq. (5) uses the current liquid or
solid mass available (either in the snowpack, if present, or in the uppermost soil level) to provide a constraint for the change of
phase rate $M_g$. If we denote the net energy flux at the surface by $B$, the temperature increment at the ground computed in the
model can be expressed as

$$\Delta T_g = -\frac{(B - M_g H_f)}{\frac{\partial B}{\partial T_g}} \tag{6}$$



where $H_f$ is the latent heat of fusion of water ($H_f = 334\,\mathrm{KJ\,Kg^{-1}}$). The new temperature $T_g + \Delta T_g$ will then be propagated downward through snowpack by solving implicitly the heat diffusion process. We denote this solution for the energy balance at the surface as "*explicit melt*", or EM, as the ground temperature tendency and an estimate of the change of phase rate $M_g$ are computed together when solving eq. (5). This is in contrast with a possible alternative simpler approach in which first the temperature profile is updated under the assumption of no change of phase taking place, and then an estimate of melting or freezing rates for each snow level is computed by "using up" the excess or deficit of energy in each snow level where temperature is above freezing and ice is present, or conversely where temperature is below freezing and liquid water is present. We denote here this second approach "*implicit melt*", or IM. While GLASS is based on the EM approach, as was the previous snow model used in LM4.1, we perform a numerical experiment to compare both approaches (see Section 5.3) and find that IM may not be viable for applications where long time steps are required, such as in global climate simulations.

### 3.3  Snowpack bulk mass and energy balance

In GLASS, the evolution of the snowpack is computed by solving the energy and mass conservation equations for each snow layer. The model representation of all physical processes is energy and mass conserving, as required for century-long simulations. The mass balance of total (liquid and frozen) water for the entire snowpack reads

$$\frac{dW_{sn}}{dt} = f_l + f_s - E_g - I_{sn,l} - R_{sn,l} - R_{sn,s} \tag{7}$$

where $W_{sn} = W_{sn,l} + W_{sn,s}\,[\mathrm{kg\,m^{-2}}]$ is the total water content of the snowpack (i.e., the snow water equivalent), and the fluxes ($[\mathrm{kg\,m^{-2}s^{-1}}]$) represent respectively liquid ($f_l$) and solid ($f_s$) effective precipitation (i.e., net of any canopy interception), the snow sublimation rate ($E_g$), the liquid water flux to the underlying soil ($I_{sn,l}$), and the liquid and solid contribution to the grid cell runoff from the snowpack ($R_{sn,l}$ and $R_{sn,s}$). If snow is present on the ground, only sublimation occurs in GLASS.

The mass balance for liquid $W_{sn,l}$ and solid $W_{sn,s}$ water for the entire snowpack read, respectively

$$\frac{dW_{sn,l}}{dt} = f_l - I_{sn,l} - R_{sn,l} + M_{tot} \tag{8}$$

and

$$\frac{dW_{sn,s}}{dt} = f_s - E_g - R_{sn,s} - M_{tot} \tag{9}$$

where $M_{tot}$ is the total melt (or freeze) rate. Within the snowpack, the local liquid water and ice balance reads:

$$\frac{dw_l}{dt} = \frac{dq_l}{dz} + M(z) \tag{10}$$

$$\frac{dw_s}{dt} = -M(z) \tag{11}$$

where $M(z)$ is local the melt (or freeze) rate at depth $z$, and $q_l$ the vertical flux of liquid water $[\mathrm{kg\,m^{-2}s^{-1}}]$. Note that, when the explicit melt is used, the total melt is computed in two steps: First, when solving implicitly the surface energy balance, an





estimate of the total melt is obtained and the corresponding latent energy flux contributes to the energy balance at the snow surface. Then, when solving the vertical water flow and surface energy balance through the snowpack, additional melt or freeze can occur in order to satisfy thermodynamic constrains (i.e., the thermodynamic equilibrium of each layer is computed, which can lead to additional melt or freeze occurring).

The vertical energy balance in the snowpack is expressed as

$$\frac{\partial H_{sn}}{\partial t} = \frac{\partial q_h}{\partial z} \tag{12}$$

with $H_{sn}$ being the energy content of the snow, and $q_h$ — vertical heat flux given by

$$q_h = -\lambda \frac{\partial T}{\partial z} + c_l q_l (T_l - T_F) \tag{13}$$

With $T_l$ the temperature of the vertical water flow of rate $q_l$. The boundary condition at the top of the snow is given by the net heat flux at the surface (Eq. (5)) and reads

$$q_h|_{z=0} = R_{net} - H_g - [L_0 + c_p (T_{z=0} - T_F)]E_g + c_l (T_l - T_F) f_l \tag{14}$$

with $T_{z=0}$ and $T_{prec}$ the temperature of the snowpack top and of liquid precipitation $f_l$, respectively, and $R_{net}$ the net radiation absorbed at the surface. This balance can also be evaluated just below the snow surface

$$q_h\bigg|_{z=0} = -\lambda \left(\frac{\partial T}{\partial z}\right)\bigg|_{z=0,\ snow\ top} - c_s (T_{z=0} - T_F) E_g + L_f E_g + c_l (T_{z=0} - T_F) f_l \tag{15}$$

By combining eq. (14) and (15) we obtain

$$\lambda \left(\frac{\partial T}{\partial z}\right)\bigg|_{z=0,\ snow\ top} = R_{net} - H_g - [L_f + L_{vap} + c_p (T_{z=0} - T_F)] E_g + c_l (T_{prec} - T_{z=0}) f_l \tag{16}$$

The bottom boundary condition (at $z = z_b$) between snow and soil reads

$$-\lambda \left(\frac{\partial T}{\partial z}\right)\bigg|_{z=z_b,\ soil\ top} = \lambda \left(\frac{\partial T}{\partial z}\right)\bigg|_{z=z_b,\ snow\ bottom} + c_l (T_{l,z=z_b} - T_F) I_{sn,l} + c_s (T_{l,z=z_b} - T_F) I_{sn,s} \tag{17}$$

and additionally we impose $T_{z=z_b,\ snow\ bottom} = T_{z=z_b,\ soil\ top}$.

### 3.4 Solution of water and energy balance of the snowpack

Following the approach used in LM4.1 (Milly et al., 2014), we solve the energy conservation within the snowpack by separating "dry" (heat conduction) from "wet" processes (those related to vertical fluxes of liquid water).

The heat content of a snow layer is defined with respect to ice at freezing temperature — i.e., for snow layer $k$ the heat content is defined as

$$H_{sn,k} = (c_l w_{l,k} + c_s w_{s,k})(T_k - T_F) + H_f w_{l,k} \tag{18}$$

The vertical heat diffusion equation is solved by discretizing the equation over the vertical layer structure of the snowpack.





For layer $k$ we have

$$c_k \frac{\partial T_k}{\partial t} = H_{k-1}(T_{k-1}, T_k) - H_k(T_k, T_{k+1}) + S_k \tag{19}$$

where $H_k$ is downward heat flux through the bottom of the layer $k$ to layer $k+1$, and $S_k$ represents a heat source term in layer $k$ (e.g., from the absorption of solar radiation). We linearize fluxes around their values of the beginning of the time step:

$$H_k = H_{0,k} + \frac{\partial H_k}{\partial T_k}\Delta T_k + \frac{\partial H_k}{\partial T_{k+1}}\Delta T_{k+1} = H_{0,k} + \Lambda_k \Delta T_k - \Lambda_k \Delta T_{k+1} \tag{20}$$

At the top of the snowpack, atmosphere supplies the net heat flux as described in Section 3.2. Conversely, at the bottom of the snowpack, the heat flux is provided by the uppermost soil layer:

$$G(T_N) = G_0 + \frac{dG}{dT}\Delta T_N \tag{21}$$

### 3.5 Heat conductance between snow layers

For two consecutive snow layers, $k$ and $k+1$, each with its own thickness and its own heat conductance, the resistance to downward heat flux between layers can be expressed as

$$\frac{1}{\Lambda_k} = \frac{\Delta z_k}{2\lambda_k} + \frac{\Delta z_{k+1}}{2\lambda_{k+1}} = \frac{1}{2}\frac{\lambda_k \Delta z_{k+1} + \lambda_{k+1}\Delta z_k}{\lambda_k \lambda_{k+1}} \tag{22}$$

The downward heat flux from layer $k$ to $k+1$ is then:

$$H_k = \Lambda_k(T_k - T_{k+1}) \tag{23}$$

### 3.6 Snow thermal conductivity

The thermal conductivity of the snowpack is parameterized as a function of snow density $\rho_k$ in each layer, $(\mathrm{kg\,m^{-3}})$ using the parameterization by (Yen, 1981) used e.g., in Lafaysse et al. (2017). Effective snow heat conductance $\lambda_k$ $(\mathrm{W\,m^{-1}\,K^{-1}})$ is expressed as

$$\lambda_k = \max\left[a_\lambda\left(\frac{\rho_k}{\rho_w}\right)^{1.88}; \lambda_{\min}\right] \tag{24}$$

with $a_\lambda = 2.22$ $(\mathrm{W\,m^{-1}\,K^{-1}})$, and $\lambda_{min} = 4\times10^{-2}$ $(\mathrm{W\,m^{-1}\,K^{-1}})$. which are made to match pure ice heat conductance $\lambda_i$ with appropriate choice of $a_\lambda$. As an alternative, we implemented the snow heat conductance formulation proposed by Calonne et al. (2011), Sun et al. (1999), and Arduini et al. (2019). In this formulation, the total snow thermal conductivity is the sum of two terms: The first is a quadratic function of density representing the actual snow thermal conductivity (Calonne et al., 2011), while a second term accounts for the additional heat advected by water vapor (Sun et al., 1999):

$$\lambda_k = \lambda_{c,k} + \lambda_{wv,k} = \left(a_1\rho_k^2 - a_2\rho_k + a_3\right) + \frac{P_0}{P_{srf}}\left(b_1 - \frac{b_2}{T_k - b_3}\right) \tag{25}$$

where $a_1 = 2.5\times10^{-6}$ $\mathrm{Wm^5K^{-1}kg^{-2}}$, $a_2 = 1.23\times10^{-4}$ $\mathrm{Wm^2K^{-1}kg^{-1}}$, $a_3 = 0.024$ $\mathrm{Wm^{-1}K^{-1}}$, $b_1 = -0.06023$ $\mathrm{W\,m^{-1}\,K^{-1}}$, $b_2 = 2.5425$ $\mathrm{W\,m^{-1}}$, and $b_3 = 289.99$ K. In this application we use eq. (25) for $\lambda_k$ as it accounts for the effects of water vapor.



### 3.7 Snow sublimation

The rate of evaporation or sublimation is computed by solving the nonlinear equation for the surface energy balance, eq. (5). In the case snowpack is present, we assume that the entire water vapor flux comes from sublimation (even if liquid water is present

in the snowpack), and the sublimating ice is lost from the uppermost snow layer. In the model, the heat diffusion through the snowpack and sublimation are resolved in two different steps. However, since in reality the two phenomena occur simultaneously, the change in heat content of the top snow layer associated with sublimating snow must account for the simultaneous change in the layer's temperature due to heat diffusion. Therefore, this nonlinear interaction between heat diffusion and heat flux due to sublimating snow is accounted for by correcting the layer's temperature to ensure that energy is conserved when

both processes are considered to occur simultaneously.

### 3.8 Implicit change of phase

Melting imposes an upper limit on the temperature profile in the snowpack, since snow temperature should not exceed melting point. If the solution of the heat equation produces temperature larger than the freezing temperatures (i.e., $T_k > T_F$ for some snow layer $k$), then the excess energy required for increasing layer's temperature of the amount $\Delta T_{*,k} = T_k - T_F$ is instead

used to melt a snow mass equal to

$$\Delta w_{s,k} = \min \left[ w_{s,k}, \quad \frac{(c_s w_{s,k} + c_l w_{l,k}) \Delta T_{*,k}}{L_f} \right] \tag{26}$$

where $L_f = 334 \, \mathrm{J \, kg^{-1}}$ is the latent heat of fusion of ice, and the expression in the numerator is the specific heat of layer $k$. For mass conservation, $\Delta w_l = -\Delta w_s$. Therefore, the heat required for the phase change is $F_{m,k} = L_f \Delta w_{s,k} < 0$. The new equilibrium temperature $T_F + \Delta T_k^{(u)}$ of the snow layer is then computed by evaluating the energy conservation equation for

layer $k$:

$$(c_s w_{s,k} + c_l w_{l,k}) \Delta T_{*,k} + w_{l,k} L_f = (c_s (w_{s,k} + \Delta w_{s,k}) + c_l (w_{l,k} - \Delta w_{s,k})) \Delta T_k^{(u)} + (w_{l,k} - \Delta w_{s,k}) L_f \tag{27}$$

which can be solved for the new snow layer equilibrium temperature. Similarly, in the case of $T_k < T_F$, if $w_{l,k} > 0$, additional energy is provided to the layer by freezing the available liquid water. Again, the amount of energy is limited by the amount of liquid water available for freezing. The temperature of the melt water is then $T_F$ while for ice $T_k < 0$. Thus, in the case of

freezing we have $\Delta T_k^u < 0$ and $F_{m,k} = L_f \Delta w_{l,k} > 0$.

This implicit formulation suffers from the underestimation of melting, as it is assumed that the temperature difference remains constant during the time step. This issue is greatly mitigated in GLASS by adopting the explicit melt scheme described in section 3.2, and the implications of this choice will be examined below.

### 3.9 Snowfall and snowpack solid water balance

In the case of snowfall, if (i) there is already snow on the ground and (ii) the new depth $\Delta z_{fall}$ is smaller than a threshold (set to half the depth of the uppermost snow layer before the frozen precipitation event), then the new snow is added to the existing





layer and no additional layers are created. The density of fresh snow is used to compute $\Delta z_{fall}$ and this snow depth is summed to the thickness of the existing layer, so that the resulting density of the merged layer will be a weighted average of those of new and existing snow (note, layer density is not a model variable and it is computed from as $w_s/\Delta z$ whenever needed). If

instead there is no initial snow on the ground, or if the amount of new snow is larger than the set threshold, a number of new snow layers are created as discussed in Section 3.1. The properties assigned to the freshly fallen snow are computed following Vionnet et al. (2012): The density $\rho_{fall}$ of new snow is

$$\rho_{fall} = a_\rho + b_\rho \left(T_a - T_F\right) + c_\rho \overline{U}^{1/2} \tag{28}$$

A three-parameters expression in which fresh snow density is expressed as a function of the mean wind velocity ($\overline{U}$), and the

atmospheric temperature ($T_a$), $T_F = 273.15 \, \text{K}$ the freezing point of water. Parameters used in eq. (28) are $a_\rho = 109 \, \text{kg m}^{-3}$, $b_\rho = 6 \, \text{kg m}^{-3} \, \text{K}^{-1}$, and $c_\rho = 26 \, \text{kg m}^{-7/2} \, \text{s}^{-1/2}$, setting a minimum snow density of $50 \, \text{kg m}^{-3}$. The temperature of the fresh snow is equal to that of precipitation, which in the current model configuration is set equal to the temperature of the lower atmosphere. The optical diameter of fresh snow is set to the constant value of $d_{opt,fall} = 10^{-4} \, [\text{m}]$, as recommended by Carmagnola et al. (2014). The (dimensionless) sphericity of fresh snow is computed as done by Vionnet et al. (2012):

$$s_{fall} = \min\left[\max\left(0.08\overline{U} + 0.38, \, 0.5\right), \, 0.9\right] \tag{29}$$

where $\overline{U}$ is the mean wind speed ($[\text{m s}^{-1}]$). Once density is known, the newly fallen snow depth can be computed as $\Delta z_{fall} = \Delta w_s / \rho_{fall}$, where $\Delta w_s = f_{prec} \cdot \Delta t \, [\text{kg m}^{-2}]$.

### 3.10 Rainfall and snowpack liquid balance

After updating the snow temperature profile and performing the solid mass balance, the mass balance for the liquid phase is

performed. In this stage, the liquid water balance is evaluated sequentially for all snow layers from the top of the snowpack down, coupled with energy conservation to determine any changes in water phase and temperature originating from the vertical water flow. For the top layer, liquid precipitation is added to the layer. Then, the new thermal equilibrium of the snow layer is computed, determining the new layer temperature and the new mass of liquid and solid water. Finally, the new solid phase properties are used to determine the pore space available for liquid water within the ice matrix of the layer. As done by Vionnet

et al. (2012), the maximum water holding capacity in each layer $W_{liq,max,k}$ is set to:

$$W_{liq,max,k} = 0.05\rho_w \Delta z_k \left(1 - \frac{\rho_{s,k}}{\rho_i}\right) \tag{30}$$

with $\rho_{s,k}$ the density of the snow layer (solid phase only), and $\rho_i = 917 \, \text{kg m}^{-2}$ the density of ice, and $\rho_w$ the density of liquid water.

### 3.11 Snow metamorphism

The snow microstructure in each snow layer $k$ is characterized by three parameters (the layer index $k$ will be omitted in the rest of this section for simplicity): Snow optical diameter $d_{opt}$, snow dendricity $\delta$, and snow sphericity $s_p$. The optical diameter $d_{opt}$





represents the diameter of a monodisperse set of spheres with the same surface/mass ratio, or Surface Specific Area (SSA). SSA can be obtained from $d_{opt}$ as $\text{SSA} = 6/(d_{opt} \cdot \rho_{ice})$, with $\rho_{ice} = 917 \, \text{kg m}^{-3}$ the density of ice. We choose to use the $d_{opt}$ as prognostic variable because, as pointed out by Carmagnola et al. (2014), it can be directly used to parameterize snow albedo. However, the optical shape of snow grains also can have significant impact on the optical properties of the medium (He et al., 2017; Robledano et al., 2023). The evolution of snow microphysical properties here is obtained as a combinations of the parameterizations proposed by Brun et al. (1992); Carmagnola et al. (2014) and Flanner and Zender (2006). In GLASS, all three snow grain properties (grain dendriticy, grain sphericity and optical diameter) are prognostic variables.

The parameterization of Flanner and Zender (2006), termed F06 in the following, is used to model the effects of dry snow metamorphism on $d_{opt}$. In this formulation, the time evolution of the snow optical diameter is computed as a function of snowpack temperature, vertical temperature gradient, and snow density. A parametric equation is used to predict the rate of change of snow effective radius

$$\frac{dr}{dt} = \left(\frac{dr}{dt}\right)\Big|_{t=0} \left(\frac{\tau}{r - r_0 + \tau}\right)^{1/\kappa} \tag{31}$$

where $\left(\frac{dr}{dt}\right)\big|_{t=0}$, $\tau$ and $\kappa$ are parameters derived from a look-up table as functions of snow density, temperature, and temperature gradient. The optical diameter of fresh snow is taken to be that corresponding to SSA of $60 \, \text{m}^2/\text{kg}$. This equation predicts an effective radius in $\mu$m which is then converted to optical diameter $d_{opt}$ in [m] used in GLASS. For wet snowpack, the additional size increase of snow grains due to wet metamorphism is described using the model put forward by Brun et al. (1992), in which grain size evolution is given by

$$\frac{dr}{dt} = \frac{10^{18} C_1 \theta_B^3}{4 \pi r_0} \tag{32}$$

where $C_1 = 4.22 \times 10^{-13}$ and $\theta_B = w_l/(w_s + w_l)$ the snow liquid fraction. Brun's equation here is expressed in terms of snow grain radius in [$\mu$m] which in our model is then converted to snow optical diameter in [m]. In addition to the optical diameter, snowpack layers are characterized by two additional parameters describing the shape of snow grains: dendricity $\delta$ and sphericity $s_p$. These are both dimensionless quantities ranging from 0 to 1. Fresh fallen snow is assumed to be in a dendritic state, with a dendricity value decreasing over time due to the combined effects of wind drift and metamorphism. When the dendricity parameter approaches zero, the snow reaches a non-dendritic state. Similarly, $s_p = 1$ indicate perfectly spherical particles and $s_p = 0$ completely non-spherical particles, i.e., faceted snow crystals.

The evolution of $s_p$ and $\delta$ due to wet and dry snow metamorphism are described according to the model by Brun et al. (1992). In the case of wet snow metamorphism, we have

$$\frac{d\delta}{dt} = -\frac{1}{16}\theta^3 \tag{33}$$

$$\frac{ds_p}{dt} = \frac{1}{16}\theta^3 \tag{34}$$





With $\theta = 100 \cdot w_l/(w_l + w_s)$ the percent liquid fraction. For dry metamorphism, the evolution equations for $s_p$ and $\delta$ are, in case of a mild temperature gradient ($G = |dT/dz| \leq 5\,\mathrm{K/m}$)

$$\frac{ds_p}{dt} = 10^9\, e^{-6000/T} \tag{35}$$


$$\frac{d\delta}{dt} = -2 \cdot 10^8 e^{-6000/T} \tag{36}$$

While in the case of an intermediate or steep temperature gradient ($G = |dT/dz| > 5\,\mathrm{K/m}$)

$$\frac{ds_p}{dt} = -2 \cdot 10^8 e^{-6000/T} G^{0.4} \tag{37}$$

$$\frac{d\delta}{dt} = -2 \cdot 10^8 e^{-6000/T} G^{0.4} \tag{38}$$

These equations hold for the case of dendritic snow. When the snow reaches a non-dendritic state, dendricity remains zero while sphericity continue evolving in time, and snow effective radius also evolves according to eqns. (31) and (32).

### 3.12 GLASS snow albedo model

In GLASS, in addition to the BRDF albedo model we employ the albedo parameterization proposed by (He et al., 2018b). In
this formulation, snow albedo is expressed as a function of snow grain shape and size as

$$\alpha_{VIS,NIR} = b_0\left(\delta_p, s_p\right) + b_1\left(\delta_p, s_p\right) R_n + b_2\left(\delta_p, s_p\right) R_n^2 - \Delta\alpha_{IMP} \tag{39}$$

where

$$R_n = \log_{10}\left(\frac{R_e}{R_0}\right) \tag{40}$$

with $R_e$ the snow grain effective radius ($R_e = 3V_s/(4A_s)$).This is related to SSA as $R_{SSA} = 3/\rho_i/d_{opt}$. For convex shapes
$R_{SSA} = 1$ while for Koch snowflake $R_{SSA} = 0.544 R_e$. The correction term $\Delta\alpha_{IMP}$ accounts for the effect of impurities deposited on old snow. While this phenomenon will be examined separately in future extensions of this study, here we use the simple correction used in CROCUS and obtained for an alpine site conditions ($\Delta\alpha_{IMP} = age/60$) In the case of direct radiation, the dependence of snow albedo on the direction of incident radiation is accounted for following Marshall (1989)

$$R_e' = R_e\left(1 + a_b \Delta\mu\right) \tag{41}$$

with $a_b = 0.781$ for visible band ($b = VIS$) and $a_b = 0.791$ for NIR band ($b = NIR$). For diffuse radiation, we set $\mu_D = 0.65$ which corresponds to $\theta = 49.5°$.

Grain shape has been recognized to play an important role in determining the optical properties of the snow medium (Robledano et al., 2023). He et al. (2018a) developed the parameterization defined by eq. (39) for different snow grain shapes,





idealizing snow as a collection of either (a) spheres, (b) spheroids, (c) hexagons, or (d) Koch snowflakes. In GLASS, snow
microphysical properties are represented through three variables: grain sphericity, dendricity and optical diameter. These three
parameters are used to characterize the effect of snow grain shape on snow reflectivity. In particular, a grain size with den-
dricity larger than $0.5$ is considered for radiative balance purpose as a Koch snowflake. Conversely, non-dendritic or weakly
dendritic snow is modelled as a collection of spheroids, hexagonal crystals, or spherical particles. For high-sphericity parame-
ter ($s_p > 0.8$), we compute snow albedo using the parameters relative to a collection of spheres in eq. (39). For non-spherical
or weakly spherical snow, the parameters relative to a collection of hexagons are used. In the remaining case (non dendritic
snow, with sphericity larger than 0.2 but smaller than 0.8) the spheroid case is used. This approach allows us to capture the
effect of snow grain shape on the optical properties of the snowpack. To our knowledge, this is the first time a snow model
developed for Earth System model simulations includes a prognostic description of snow grain shape and its effect on snow
optical properties. Numerical studies have shown that accounting for shape can impact snow optical properties depending on
snow optical diameter and content of impurities (He et al., 2017).

In GLASS, when snow is thick enough, shortwave radiation penetrates the snowpack. The absorbed radiation is distributed
exponentially within the snowpack if this is thick enough ($d > 0.02\,\mathrm{m}$) as

$$Q_s\left(z\right) = \sum_{b=1}^{2} \left(1 - \alpha_b\right) R_{s,b} e^{-\beta_b z} \tag{42}$$

The penetration of light in the snow is evaluated as in CROCUS for our two bands (Visible and NIR). For visible light,
$\beta_{VIS} = 0.003759\,\rho\,d_{opt}^{-0.5}$), with density and optical diameter averaged over the near-surface layer of the snowpack, up to
3cm. For the NIR band, $\beta_{NIR} = 400$. These values follow the values proposed byJordan (1991) and by Shrestha et al. (2010).

### 3.13 Models steps summary

Figure 2 provides a schematic representation of the computational steps performed to update the state of the snowpack at
each model time step. Due to the nature of the implicit solution adopted for the energy and water balance, the heat diffusion
through the snowpack must be solved in two separate steps. First ("*step 1*"), the heat fluxes through the snowpack are computed
starting from the lower boundary accounting for possible heat sources within the snowpack (e.g., due to shortwave radiation
absorption). In this first step, an estimate of the ice available for melting is also computed. Then, the surface energy balance is
performed, according to eq. 5. Solving this equation yields the tendency for the surface temperature $\Delta T_g$ as well as the amount
$M_g$ of melting ice or freezing water, depending on its sign. This information is then used in the second model step ("*step 2*") of
the snow energy and mass balance: The temperature profile in the snow is first updated based on the upper boundary tendency
$\Delta T_g$ and the vertical fluxes obtained in step 1. The mass of liquid and ice in the snowpack is then updated, based on the
estimate of water changing phase ($M_g$) previously computed. Note that after this step ("explicit melt") it is still possible that
the solution of the heat equation yields above-freezing temperatures in some snow layers, or below-freezing temperatures in
layers where liquid water is present, which are resolved with an additional change of phase. This second "implicit melt" is then
applied by evaluating the thermal equilibrium of each snow layer: In the case of layers with solid ice and temperature above
freezing, a new equilibrium temperature is computed and the excess heat is used to melt part of the available ice. Conversely,





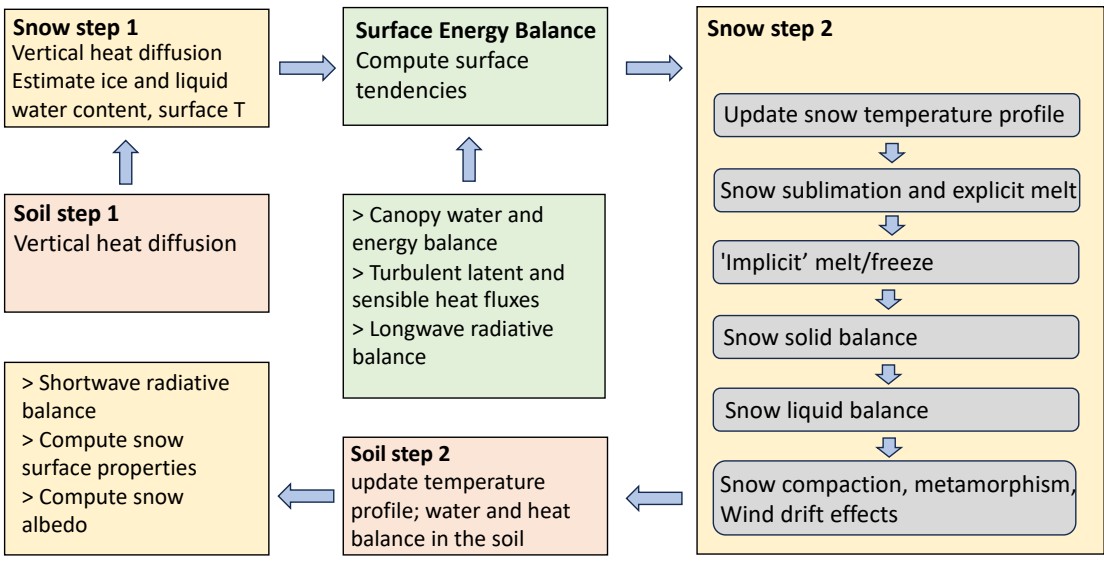

**Figure 2.** A schematic representation of the two main model steps in GLASS ("step 1", "step 2", and snow radiative properties) and their interface to the other physical processes in the GFDL LM4.1.

in case of layers containing liquid water and below-freezing temperature, liquid water is frozen until thermal equilibrium is reached.

After performing the energy balance, the following steps are computed. Fresh snow is added to the snowpack if snowfall is present. The model first tries to add the new snow to the existing uppermost layer. If the snowfall mass exceeds a threshold, a variable number of layers is added to the top of the snowpack. The properties of fresh snowfall are computed based on environmental variables (e.g., sphericity, dendriticy and density). We then perform the liquid balance in the snowpack: Water from liquid precipitation is added to the top layer. The maximum liquid water capacity of the layer is computed as a fraction of the layer pore space, given by eq. (30). If this liquid water content is exceeded, the excess water flows vertically to the underlying layers. We then solve sequentially the water balance for all layer, down to the bottom of the snowpack. We then perform in sequence snow compaction (described in Appendix B), wet and dry snow metamorphism, and evaluate the effect of wind drift (see Appendix A). Finally, we re-layer the snowpack, trying to merge or split layers based on their distance from the optimal layering structure for the given snow depth, as discussed in Section 3.1. This concludes the snow time step. After that, the integral properties of the near-surface snow layer are computed. The near-surface layer is defined as the top three

 

**Table 2.** Experimental sites used for model validation.

| Station | ID | Obs. Years | Lat. | Lon. | Elev. | Climate | Veget. |
|---|---|---|---|---|---|---|---|
| Col De Porte | CDP | 1994-2014 | 45.30 N | 5.77 E | 1325 m | Alpine | No |
| Reynolds M. E., USA | RME | 1988-2008 | 43.06 N | 116.75 W | 2060 m | Alpine | " |
| Senator Beck, USA | SNB | 2005-2015 | 37.91 N | 107.73 W | 3714 m | Alpine | " |
| Swamp Angel, USA | SWA | 2005-2015 | 37.91 N | 107.71 W | 3371 m | Alpine | " |
| Weissfluhjoch, CH | WFJ | 1996-2016 | 46.83 N | 9.81 E | 2540 m | Alpine | " |
| Sapporo, JP | SAP | 2005-2015 | 43.08 N | 141.34 E | 15 m | Maritime | " |
| Sodankyla, FI | SOD | 2007-2014 | 67.37 N | 26.63 E | 179 m | Arctic | " |
| Old Jack Pine, CA | OJP | 1997-2010 | 53.92 N | 104.69 W | 579 m | Boreal | Pine |
| Old Aspen Site, CA | OAS | 1997-2010 | 53.63 N | 106.20 W | 600 m | Boreal | Aspen |
| Old black Spruce, CA | OBS | 1997-2010 | 53.99 N | 105.12 W | 629 m | Boreal | Spruce |

centimeters or the entire snowpack, whatever is smaller. These properties will be used to compute snow albedo as discussed in Section 3.12.

# 4 Data and methods

## 4.1 Forcing and validation data

To test model performance, we employ a reference dataset widely used in the snow modelling community (Ménard et al., 2019),
and used — for example — in SnowMIP project (Krinner et al., 2018; Menard et al., 2021). Details for each observation site are reported in Table 2 and the location of the sites is shown in Figure 3. The 10 sites span a range of climates, elevation and terrain types. In particular, three of the sites are forested while the other 7 are characterized by either bare soil, or grass and low vegetation. The three forested sites located in the Canadian boreal forest are described in (Bartlett et al., 2006).

For this reason, for the 7 sites with little to no vegetation we run the model turning off vegetation. For the three forested
sites, the long spin-up allows model vegetation to fully develop before starting the historical run with in-situ meteorological forcing. We force two of the sites (*ojb, obs*) to grow evergreen vegetation, while for the third (*oas*) we force deciduous species only, to match the existing vegetation types.

In this dataset, each site includes both in-situ meteorological forcing for the observation period (as reported in Table 2), and a locally-corrected GSWP3 forcing dataset for the period 1980-2015. This forcing dataset is used here to spinup the model
for each station up to the year when in-situ metereological observational record begin. After that point, the experiment is run forcing the model with in-situ data. Both GSWP3 and in-situ forcing data are at hourly temporal resolution, and are interpolated at the model 30-min time step. Atmospheric forcing input to LM 4.1 includes liquid and frozen precipitation, downward radiation (direct and diffuse, for both visible and near-infrared bands), longwave radiation, air temperature, humidity, pressure



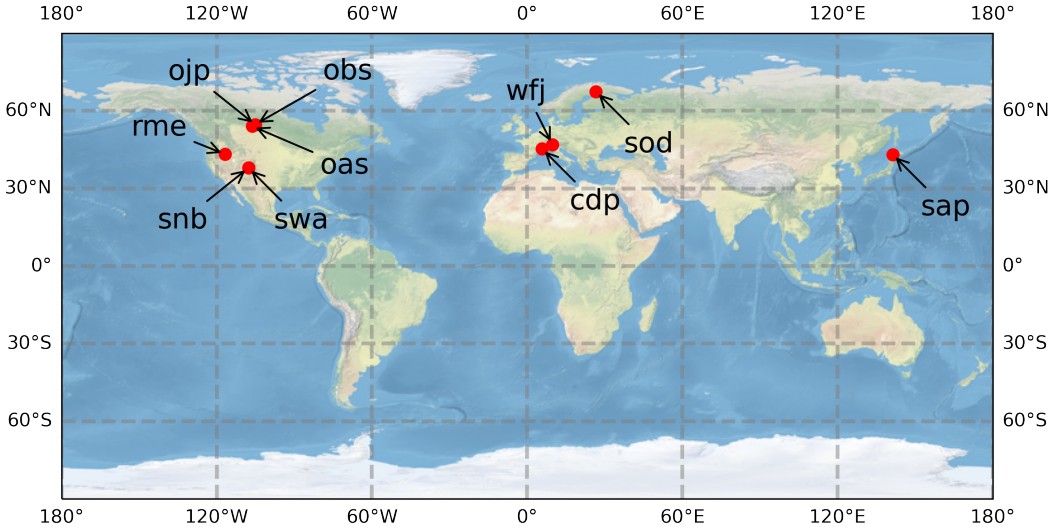

**Figure 3.** Sites used for model validation. Information about the sites are reported in Table 2.

and wind speed. For several sites (except, for example, Col de Porte site), the change in effective instrument height due to
the presence of the snowpack is not accounted for. The dataset forcing only provides total downward shortwave radiation: In
our experimental setup, this was divided between direct and diffuse (0.46 and 0.54 of the total flux, respectively) as well as
in visible and near infrared bands (0.41 and 0.59 respectively) using average climatological values obtained from the GFDL
AM4.0 (Zhao et al., 2018) atmospheric model output.

## 4.2    Experimental setup

For all sites, we spinup the land model, evaluating soil temperature and soil ice content. We found that for all sites presented
here in our model the soil is not frozen during the summer, so that the long equilibration times characteristic for permafrost
regions are not an issue, and therefore for sites with little vegetation equilibrium is reached after less than 20 model years.

For model spinup, we use at each study site the corrected GSWP3 data provided by Ménard et al. (2019). The model spinup
runs for 200 years from 1781 to 1981, cycling through the forcing for the decade 1981-1991. This allows for the soil to
equilibrate and for vegetation to grow in the sites where it is present (i.e., the three Canadian BERMS sites).

After the spinup, the model is run using GSWP3c forcing data up to the date when in-situ forcing measurement start, which
for each site is shown in Table 2. Then, the actual experiment is run for the entire length of the in-situ forcing dataset.

## 4.3    Performance Metrics

For a description of statistical model performance metrics, we follow Lafaysse et al. (2017). To assess the improvement in
model performance due the new snow scheme we compute a set of goodness of fit measures. For model configuration $i$ (e.g.,





$i$ =GLASS or $i$ =CM) we compute bias and root mean square error as

$$\hat{B}_i = \frac{1}{N_i} \sum_{k=1}^{N_i} (m_{k_i} - o_k) \tag{43}$$

and

$$\hat{R}_i = \left[ \frac{1}{N_i} \sum_{k=1}^{N_i} (m_{k_i} - o_k)^2 \right]^{0.5} \tag{44}$$

where $N_i$ model simulated values ($m_{k_i}$) are compared to $N_i$ observed values ($o_k$).

# 5   Results

## 5.1   Bulk snow properties

Simulating the seasonal evolution of snowpack over the 10 SnowMIP sites allows to demonstrate the behavior of the LM-GLASS model over a wide range of climate conditions. Time series of snow water equivalent (defined here in $\mathrm{kg\,m^{-2}}$ of snow)
is reported in Figure 4 for all sites. For each site, we show the last 6 snow-years of the simulation, including as a benchmark the results from old snow model LM-CM. Overall, both models appear in good agreement with observations for most stations and simulated snow-years. Depending on the station, LM-GLASS produces SWE estimates that are either very similar to the old LM-CM snow scheme, or larger in magnitude. The latter is the case for the *swa*, *snb* and *wfj* sites. In these cases, the larger SWE magnitudes predicted by LM-GLASS are closer to observations. These differences in SWE predictions primarily
originate from difference in modelled snow optical properties, with predicted broadband albedo values that tend to be lower for the LM-CM model. For example, for the *swa* site with some of the largest differences between the two models, the BRDF albedo scheme used in LM-CM leads to a significant underestimation of daily albedo (Fig. 5A). For the three BERMS forested sites (*ojp*, *obs*, and *oas*), where the model simulates the effects of multi-layer canopy on radiative fluxes, the SWE predictions of the two models are much closer (Fig. 5B). However, in this case modelled and observed albedo values differ significantly.
Arguably, this behavior primarily originates due to the fact that the vegetation structure produced for the model at this site does not match the dense canopy of the experimental site.

When comparing snow depths predicted by the two models, again LM-GLASS yields generally larger snow depths compared to the LM-CM (Figure 6). This behavior is not limited to the sites characterized by appreciable SWE differences between the two models. For instance, in the case of the three BERMS forested sites, LM-GLASS predicts thicker snowpack despite
predicting virtually the same SWE, indicating lower snow density than the constant value used in LM-CM.

## 5.2   Model performance metrics over the SnowMIP sites

To quantify the relative performance of the two model configurations, we compute for all sites the metrics introduced in section 4.3, namely bias and RMSE. To attribute the change in model performance to revised snow optical properties, or to other snow







**Figure 4.** Comparison in predicted SWE between LM-CM (orange) and LM-GLASS (blue). Manual observations (and automatic observations, for sites where these are available) are reported by black markers.



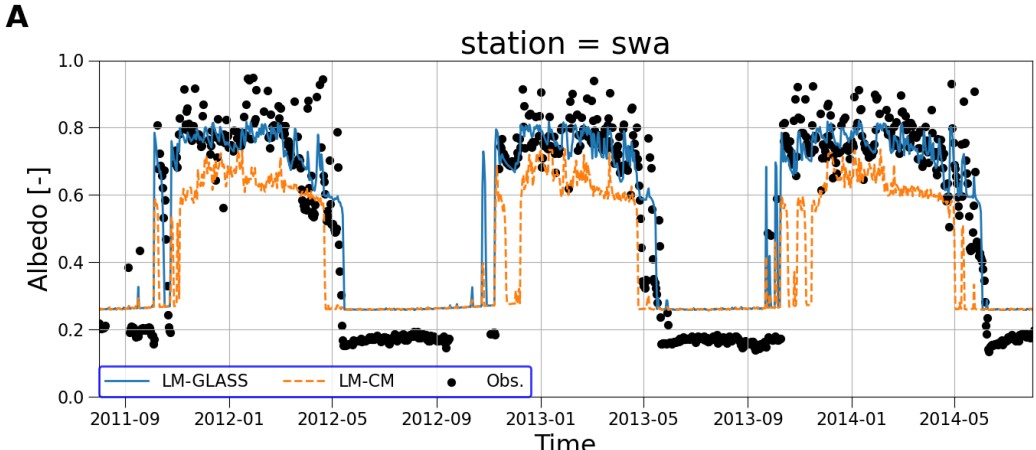

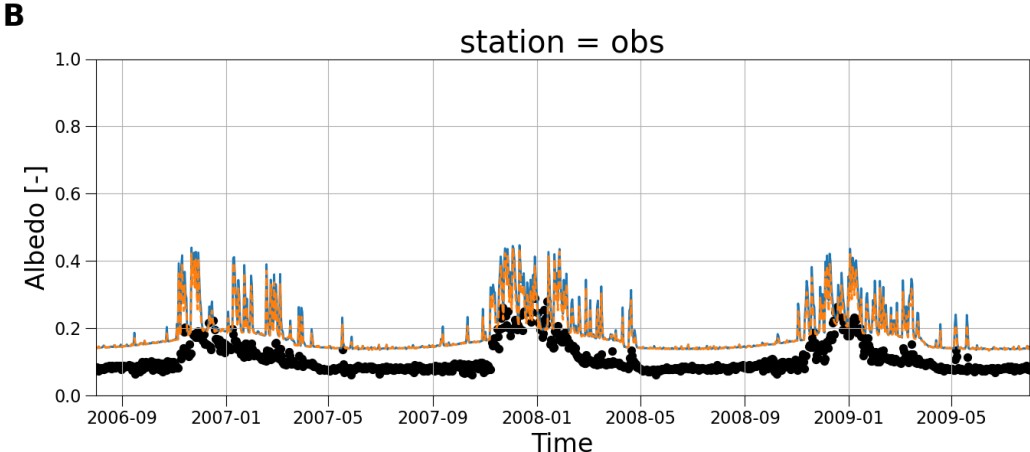

**Figure 5.** Comparison in predicted daily average broadband albedo for the Swamp Angel (swa) site (panel A) and the BERMS Old Black Spruce (obs) site (panel B). Model results are shown for LM-CM (orange dashed lines) and LM-GLASS (blue solid lines). Daily albedo observations are reported by black markers.





**Figure 6.** Comparison in predicted snow depth between LM-CM (orange) and LM-GLASS (blue). Manual observations (and automatic observations, for sites where these are available) are reported by black markers.





physical properties, we compare the old snow scheme (LM-CM) not only to the new snow model (LM-GLASS), but also

to a version of GLASS in which the original BRDF albedo model is retained (LM-GLASS-BRDF). Error metrics for these three model configurations were computed for daily SWE, snow depth, surface albedo soil temperature, and surface snow temperature, for all sites where observations of these variables were available.

The LM-GLASS-BRDF model produces little improvement in SWE compared to LM-CM. A general underestimation of the snow amount is observed for both models at all sites. On the other hand, this bias is generally reduced in LM-GLASS

simulations, leading to a substantial improvement in SWE estimates as shown in Figure 7A.

When considering predictions of snow depth, the difference in performance between the models is less marked compared to SWE results (Figure 7B). LM-CM generally exhibit a small overestimation, which is mitigated in the case of LM-GLASS-BRDF. In the case of LM-CM, this result implies that the snow density in the model, which is constant, is lower than observations, given the underestimation observed for SWE. When the full LM-GLASS model is considered, multiple sites show

modest positive model biases in snow depth. Since SWE predicted by LM-GLASS is the best between the three model configurations, these discrepancies in snow depth also arise from an underestimation in snow density, which in LM-GLASS is primarily driven by the process of snow compaction described in Appendix B.

For daily snow albedo (Figure 7C) LM-GLASS performs better than both LM-GLASS-BRDF and LM-CM, which generally underestimate daily albedo. An exception to this behavior is observed for two of the BERMS sites, where all models signifi-

cantly overestimate surface albedo. We argue this is a consequence of the land model failure to correctly represent vegetation characteristics and snow-vegetation interactions at these sites, as already shown in the case of albedo at the *obs* site, reported in Figure 5B. The effect of the revised albedo model in LM-GLASS is negligible at these forested sites, where a large role is played by the energy balance of canopy layers above the snow. Differences in predicted albedo are very small between LM-CM and LM-GLASS-BRDF. This is not surprising since these two snow scheme have an identical BRDF albedo specifications,

and any differences between the two models thus arise as a consequence of different snow surface temperature values, which is the only snowpack property used in the BRDF albedo parameterization.

Finally, biases in snow surface temperature are reported in Figure 7D for the sites where this variable was recorded. For this variable the new LM-GLASS model exhibit a larger negative bias compared to LM-CM. Since the bias is also smaller for LM-GLASS-BRDF than for LM-GLASS, this behavior is at least in part connected to the larger surface reflectivity, which

however does agree with daily albedo observations as shown in Figure 7C. As an example, three snow-years of daily surface temperature from model simulations and observations for the Col De Porte site are shown in Figure 8. Here it can be seen that while overall temperature variations are consistent between models and observations, cold temperature extremes during winter are several degrees lower in the case of LM-GLASS.

For all these snowpack variables, RMSE was also computed to complement bias, and is reported in Figure 9.

When examining error metrics in soil temperature, there are important differences between LM-CM and LM-GLASS. LM-CM exhibits a consistent negative bias at all sites examined here, of up to $-2.5\,\mathrm{K}$. This bias in soil temperature is greatly mitigated in LM-GLASS. This improvement in soil temperature is also observed in case of LM-GLASS-BRDF, indicating that this behavior is not directly related by the update in snow optical properties. Rather, we argue that the improvement originates





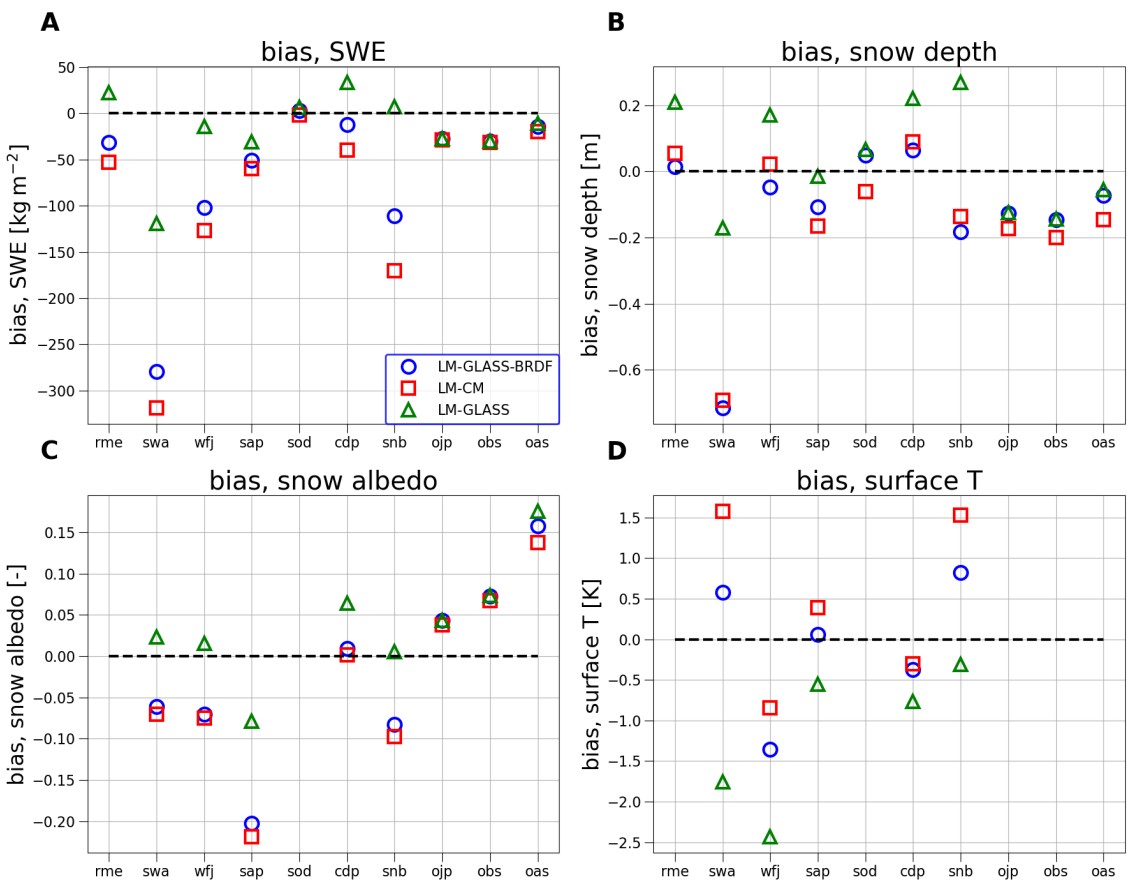

**Figure 7.** Model bias in SWE (panel A), snow depth (panel B), albedo (panel C) and surface temperature (panel D) at the 10 SnowMIP sites. Results for each variable are shown for all sites where observations are available.





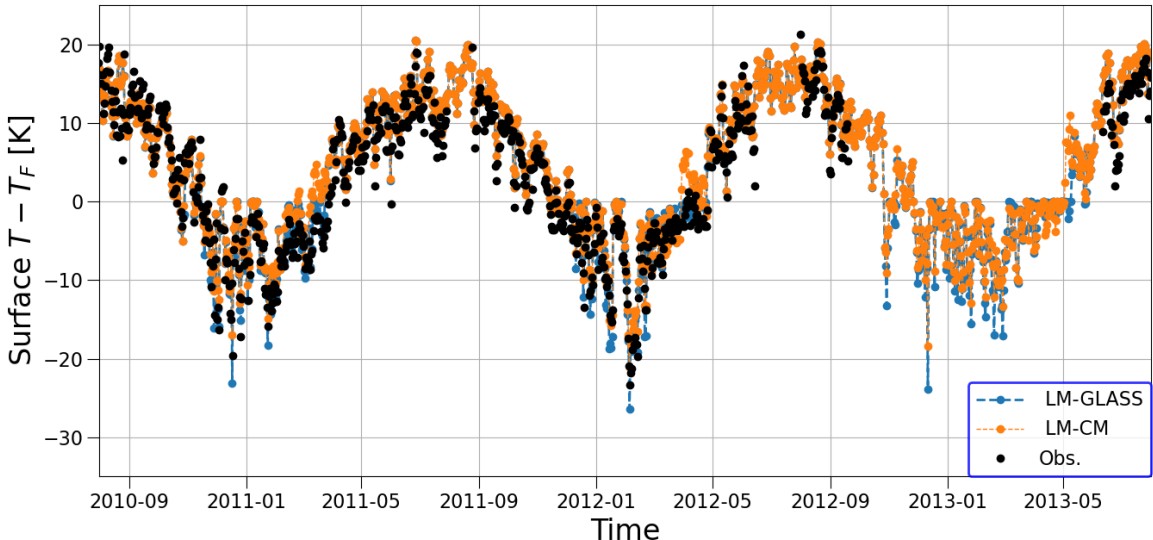

**Figure 8.** Comparison of predicted daily average surface temperature between the LM-CM (orange dashed line) and LM-GLASS (blue solid line) models for the Col De Porte site. Observed values (daily averages) are reported as black markers.

in a refined representation of snow heat conductance. LM-GLASS is not only characterized by a finer vertical discretization of

the snowpack, but also by the explicit representation of snow compaction. Snow heat conductance in LM-GLASS is explicitly modelled as a function of snow density. Therefore, the insulating properties of snow with respect to the underlying soil layers are more realistic in LM-GLASS, although a small negative bias remains indicating that to some extent the actual snow heat conductance could be smaller than predicted by LM-GLASS.

This results also suggest a potential reason for the colder snow surface values predicted by LM-GLASS and discussed earlier.

The near-surface snow layers in GLASS can be thinner than those in LM-CM, especially in the case of thick snowpack. In this case, it is not surprising that thin surface layers with small heat capacity and increased insulating properties of the underlying snow layers would lead to colder surface temperature. While this could be a limitation of LM-GLASS, it is also possible that cold temperatures at the surface originate from discrepancies between modelled and actual turbulent fluxes in the atmospheric surface layer.

As a representative example of the performance of the models in capturing temperature variation in the underlying soil, we show results for soil temperature at three depths for the Col De Porte site, for which observations are available (Figure 10). Modelled values are overall consistent with observations, although for the deeper layer a cold bias is observed for both models. However during the winter season when snow is on the ground the LM-GLASS predicted temperatures greatly reduce the cold bias observed for the original model LM-CM.



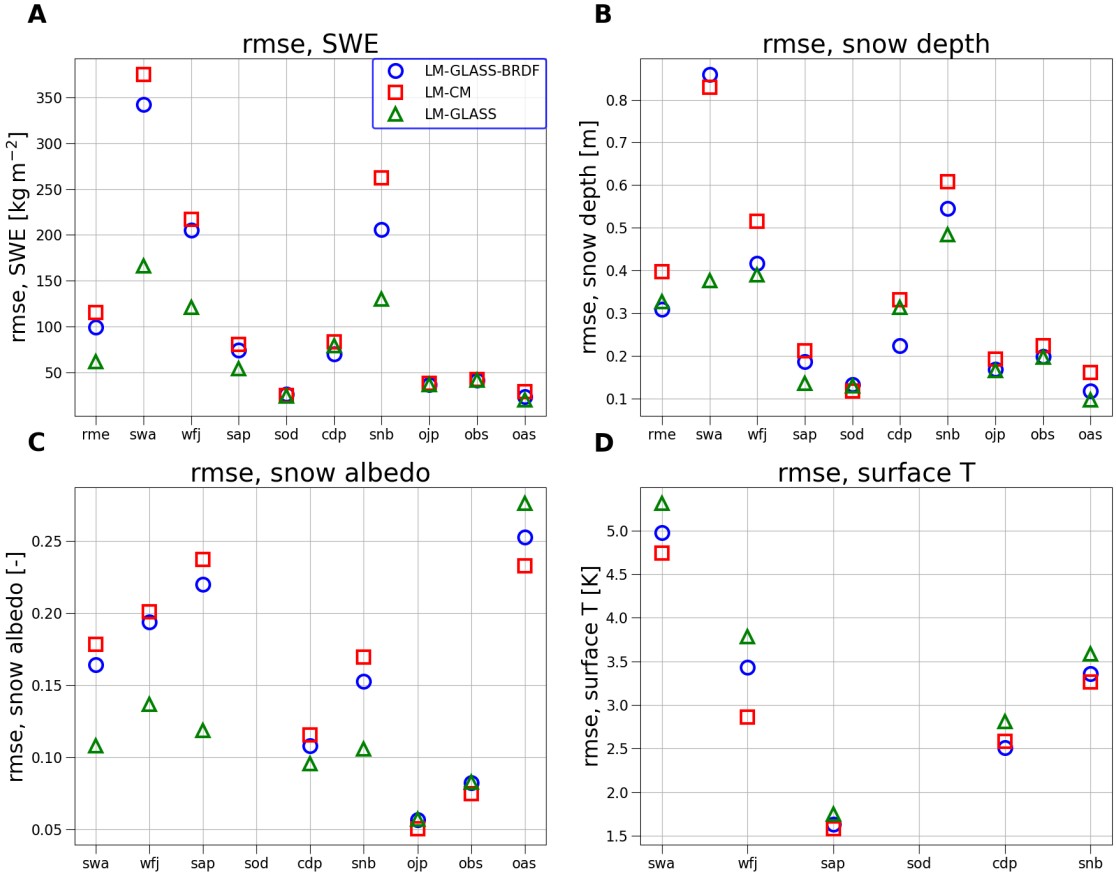

**Figure 9.** For each model configuration, rmse metrics for daily SWE (panel A), snow depth (panel B), albedo (panel C) and surface temperature (panel D) at the 10 SnowMIP sites. Results are reported for all sites where observations are available. Higher rmse values correspond to poorer model performances.



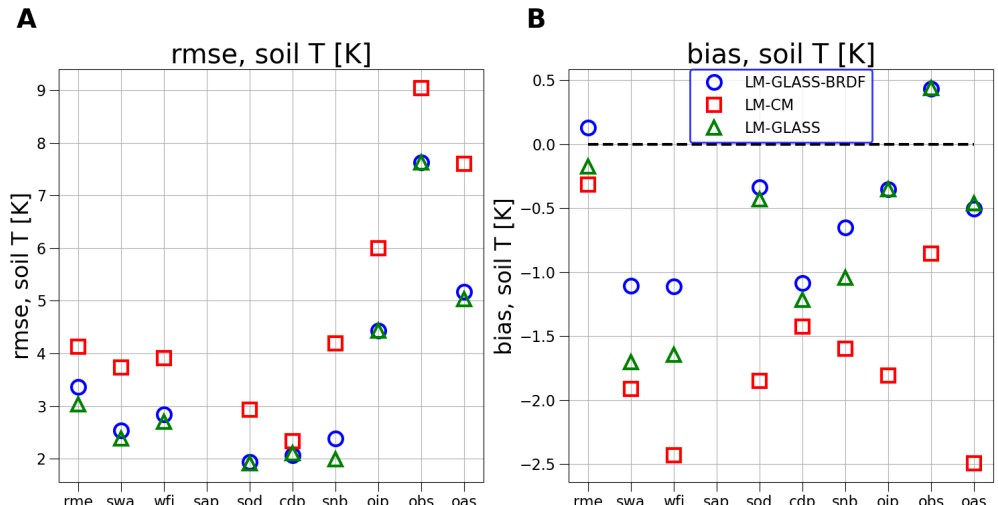

**Figure 10.** Model bias and rmse for soil temperature. For each site, error metric were computed for the uppermost soil depth at which observations were available

## 5.3 Implications of the implicit scheme to solve phase change

The treatment of melt and freeze processes can be dependent on the numerical scheme used to resolve these processes. To investigate this issue in LM-GLASS, we examine the differences of the two schemes discussed in section 3.8. SWE predictions obtained from LM-GLASS with the default explicit melt scheme are compared with the same predictions from the implicit scheme. As discussed in section 3.8, in the latter case the change of phase for each snow layer is computed only after the temperature profile is updated from solving implicitly the heat transfer through the snowpack. The differences between explicit and implicit melt schemes are shown in Figure 12 (panels A and B, respectively). In the case of implicit melt, the modelled SWE exhibits a marked dependence on the the time step used in the calculations, with longer time steps leading to an underestimation of the snow melt. On the other hand, when explicit melt is included in the model, as is the case for the default LM-GLASS configuration, this undesirable dependence on the time step effectively disappears. Within the range of time steps examined here (ranging from 5 to 30 minutes) the results of the explicit melt scheme are closer to the implicit melt results obtained for a 5 minute time step. However, while this model time resolution may be attainable for local studies, is still out of reach for global scale climate simulation. For a time step of 30 minutes, which is currently used in global simulation with GFDL LM 4.1, the difference between the two model configuration can become significant.

## 6 Discussion

A key improvement using LM-GLASS is the increase in winter soil temperature below the snowpack. It has been reported that ESMs participating in IPCC often underestimate soil temperature, especially at high latitudes (Koven et al., 2013), and





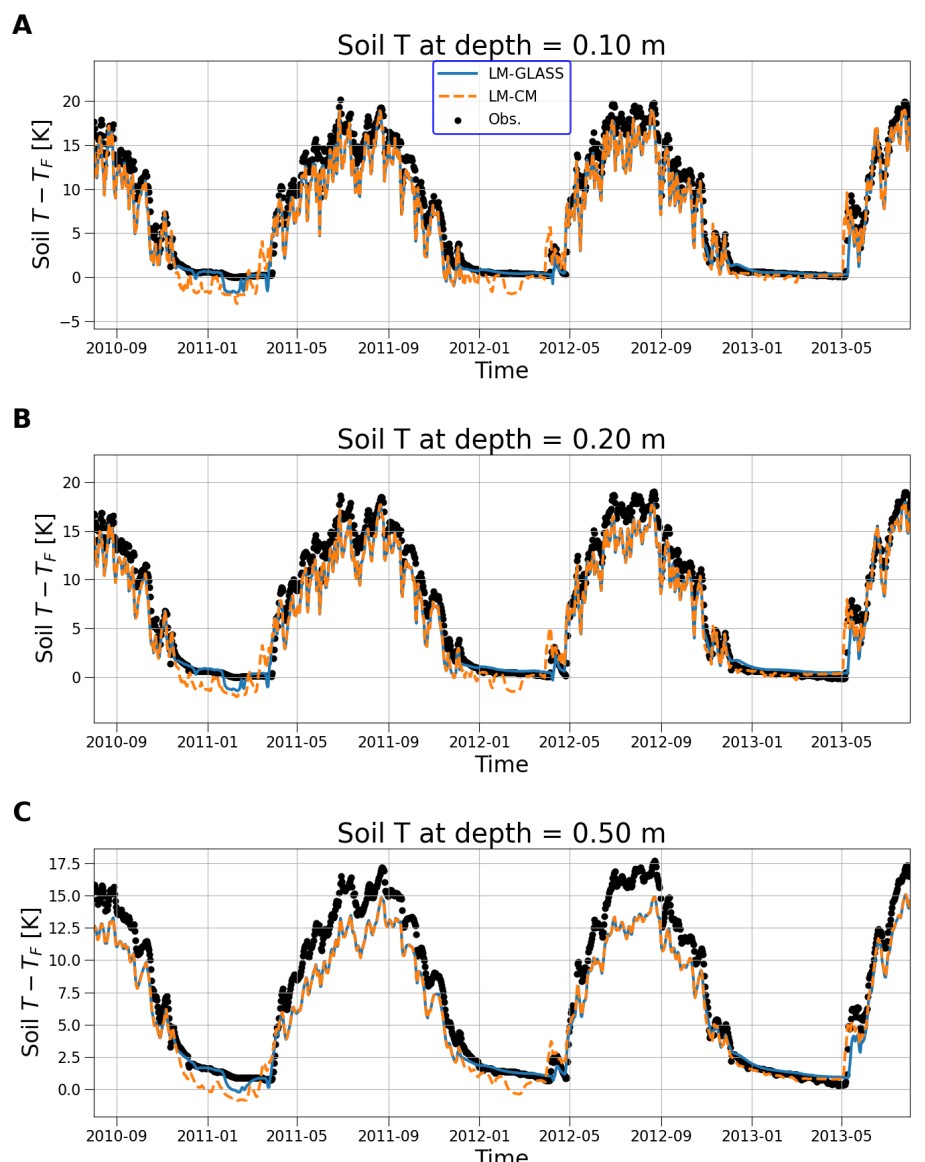

**Figure 11.** Observed and predicted soil temperature for three snow years at the Col De Porte site. Observations (black circles) and model predicted values (LM-CM as orange dashed lines, and LM-GLASS as blue solid lines) are reported at three soil depths.





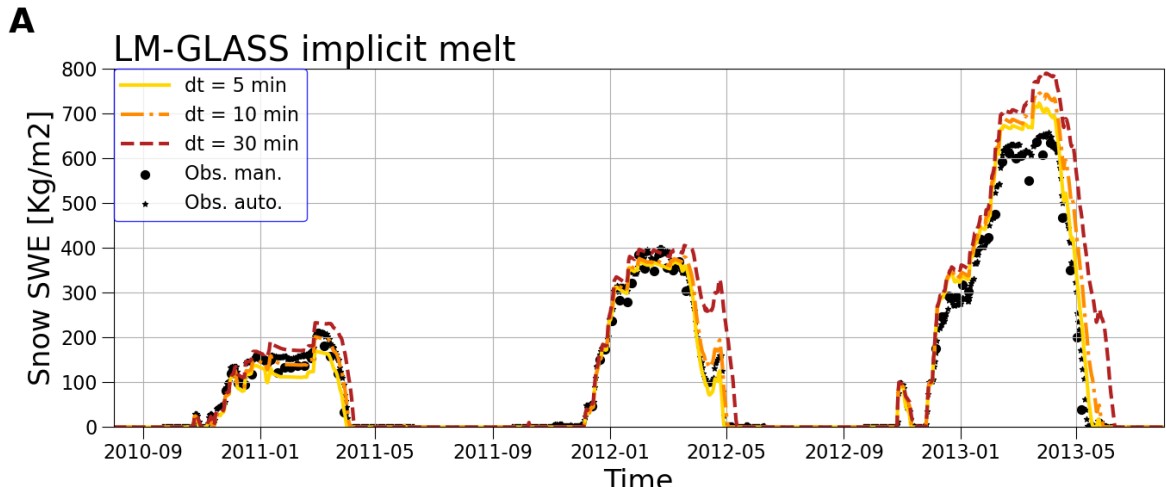

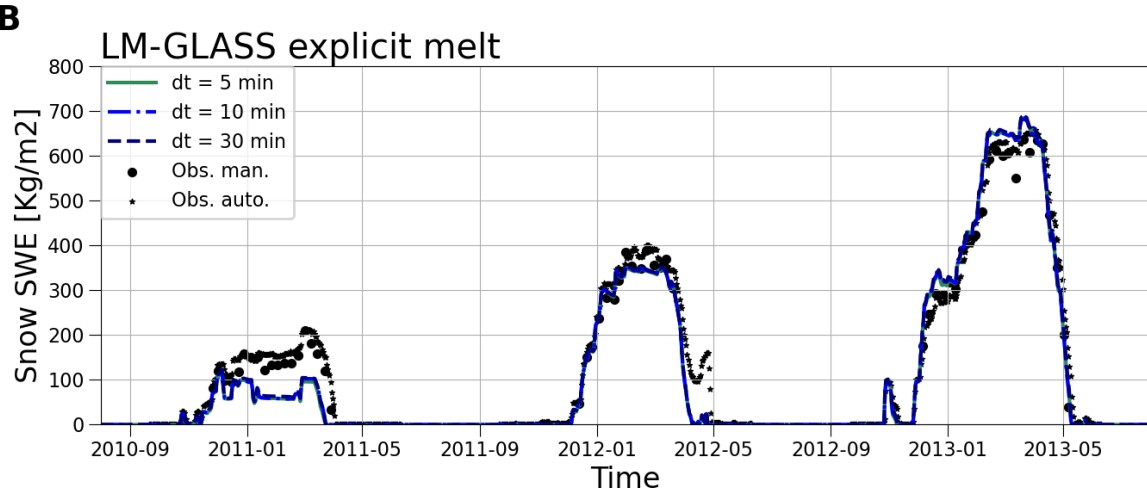

**Figure 12.** Time step dependence for the implicit melt (IM) and explicit melt (EM) numerical schemes used for snow melt. Results are shown for three snow years at Col De Porte site. Manual and automatic SWE observations are reported as black circles and star markers, respectively. Model results obtained varying the numerical time step used in LM GLASS are reported for each case. Note that in the bottom panel the results are virtually independent of the time step, so that the three lines overlap.



this is also the case for the GFDL model. Correcting this bias has important implications for the correct representation of biogeochemical cycles, as warmer soil can lead to climate feedbacks due to the release of greenhouse gases from thawing permafrost.

On the other hand, existing cold biases at the snow surface are not resolved in LM-GLASS and are even exacerbated in some cases examined in our experimental test sites. Cold temperatures observed at nighttime are common to other snow models. For example, they have been reported in the CLASS snow model (Brown et al., 2006) and are inevitably heavily dependent upon the atmospheric surface layer scheme used to compute sensible and latent heat fluxes.

     According to the classification by Boone and Etchevers (2001), the new LM-GLASS model presented here belongs to the
category of high detail snow schemes. However, the optical properties are characterized by a relatively simple parameterization (based on He et al. (2018b)) compared to the fully resolving models such as SNICAR (Flanner and Zender, 2005) or TARTES (Libois et al., 2013). Improvement in snow radiative properties are thus possible. For instance, adding to LM-GLASS the radiative effect of light absorbing impurities deposited on snow will be examined in a companion manuscript. On the other hand, the snow optical model adopted in GLASS accounts for the effect of snow grain size and shape, which was not done in
LM-CM. This development improves predictions of both daily snow albedo and SWE.

     Both snow schemes compared here (LM-GLASS and LM-CM) are based on an implicit numerical scheme. This choice is made to ensure numerical stability, given the relatively large time step ($\sim$ 30 minutes) used in global climate simulations. However, computations of the phase change within the snowpack is performed according to the so-call "explicit melt" approach, in which an estimate of the freeze/melt rate is obtained when the surface energy balance is performed. Comparing this approach
with a simpler method sometimes used in snow models (i.e., updating first the temperature profile by solving the heat diffusion through the snowpack, and only in a second step evaluating freeze or melt rates) we found that in the latter case model predictions are overly dependent on the time step used in the numerical solution. Therefore, this analysis supports the choice of the explicit melt scheme employed in both LM-CM and LM-GLASS.

     While here the model was applied at point sites so as to mimic observational datasets, the GFDL LM4.1 is routinely em-
ployed in global simulations at relatively coarse resolution (e.g., $1° \times 1°$ resolution). In such a case further research would be necessary to improve the current model with a focus on transitions between snow and no-snow areas, especially over complex terrain. Recent approaches used to model land surface heterogeneity could be useful for this purpose (Chaney et al., 2018; **?**).

     Both models compared here tend to have larger biases in the forested sites compared to the other sites, especially for daily surface albedo measured at two of the sites. This is to some extent expected as vegetation in the model is dynamic and does
not necessarily match the real canopy structure at the test sites, which can be also affected by local processes or disturbance history. Interactions of snowpack and multi-layer canopy is a known modelling challenge, and should be the focus of future research.



# 7 Conclusions

We have presented here a new snow model (LM-GLASS) tailored to the GFDL Earth System Model. While maintaining an
implicit solution for the energy balance at the snow surface, the new model provides a detailed representation of the snowpack
and its interactions with soil, vegetation and the lower atmosphere. The vertical discretization of the snowpack is now more
refined and evolves dynamically based on deposition of fresh snow, sublimation and snow melt, and performs within compu-
tational constraints. This novel vertical structure allows LM-GLASS to describe vertical variations in snow properties (snow
density, grain size and shape) which in LM-GLASS evolve dynamically in each snow layer according to the laws of snow
compaction, wind drift, wet and dry snow metamorphic processes.

Analysis of the performance of LM-GLASS and comparison with a simpler snow model (LM-CM) previously used in
the GFDL land model showed the relevance of accounting for these additional physical processes. By evaluating the model
performance over 10 SnowMIP sites, we found an overall improvement in how the model represents SWE, and comparable
results for snow depth between the two models. The observed improvement in SWE predictions primarily originates from the
updated snow optical properties , which in LM-GLASS explicitly depend on snow optical diameter and grain shape. We found
that the BRDF parameterization used in LM-CM in general tends to underestimate albedo at the test sites, and thus can lead to
overestimating snow melt.

The largest (and more consistent across sites) improvement in model skill was found for soil temperature, which was gen-
erally underestimated in LM-CM, suggesting an overestimation in heat conductance through the snowpack. In GLASS soil
temperature predictions were on average higher those obtained from LM-CM during the snow accumulation phase at all sites,
and were generally closer to observations at sites where these were collected.

The development of a refined snow model and its implementation in an Earth System Model paves the way towards multiple
future improvements and research directions. First, future work could investigate the performance of the snow scheme globally,
and in coupled land-atmosphere simulations, necessary to evaluate the strength of any feedbacks originating from the improved
realism in snow physics. LM-GLASS is expected to impact land-atmosphere interactions due to the changes in i) snow albedo,
ii) snow surface temperature, and iii) soil temperature. The latter feature is particularly encouraging, as several ESMs (including
the GFDL model) have been found to broadly underestimate soil temperature with important implications e.g., for permafrost
simulation.

Finally, future research could further improve the description of snow optical properties harnessing the vertical information
on snow grain properties provided by LM-GLASS. Towards this objective, in a companion manuscript we will present the
effects of deposition of light absorbing impurities in LM-GLASS and their effect on snow albedo and snow melt.

*Code and data availability.* The code used in this projects is made available during the review process in a zenodo repository with DOI
https://gitlab.gfdl.noaa.gov/Enrico.Zorzetto/landmod_snowmip_2022. The source code of the GLASS-LM model is available at
https://gitlab.gfdl.noaa.gov/fms/lm4P/-/tree/user/enz/jass. Note these are temporary URLs internal to GFDL, to be updated upon manuscript
acceptance.



**Appendix A: Effect of wind on snow**

The effect of wind on snow implemented in GLASS is based on the approach used in CROCUS (Brun et al., 1997; Vionnet et al., 2012) with the only difference that LM-GLASS tracks three snow variables as prognostic (snow dendricity, sphericity and optical diameter), consistent with the approach used for snow metamorphism. A mobility index parameter $M_0$ quantifies the vulnerability of a snow layer to wind erosion

$$M_0 = \begin{cases} 0.34\,(0.75\delta - 0.5s_p + 0.5) + 0.66F_\rho & \text{for dendritic snow} \\ 0.34\,(-0.583g_s - 0.833s_p + 0.833) + 0.66F_\rho & \text{for non-dendritic snow} \end{cases} \tag{A1}$$

with $F_\rho = 1.25 - 0.0042\,(\max(\rho_{\min}, \rho) - \rho_{\min})$, with $\rho_{\min} = 50\;\mathrm{kg\,m^{-3}}$.

A driftability index is then obtained as $S_l = -2.868\exp\left(-0.085\overline{U}\right) + 1 + M_0$ accounting for the effect of average wind speed $\overline{U}$, so that wind–driven snow transport occurs only for $S_l > 0$.

The effect of wind drift on snow micro structure is accounted for by evolving snow grain properties as a consequence of packing and fragmentation by the wind action. The characteristic time scale of this process for snow layer $k$ is computed as $\tau_k = \frac{\tau_{w0}}{\Gamma_k}$, with $\Gamma_k = \max(0, S_{l,k}\exp\left(-\zeta_k/0.1\right))$ and $\tau_{w0} = 48$ hours.

Here $\zeta_k$ is a depth scale in the snowpack which accounts for the hardening of overlaying layers, and reads $\zeta_k = \sum_k \left(\Delta z_k(3.25 - S_{l,k})\right)$.

The evolution of snow properties due to snow drift is computed as in CROCUS. LM-GLASS evolves sphericity, dendricity and optical diameter of snow. Additionally, snow density also evolves according to the equation

$$\frac{d\rho}{dt} = \frac{\rho_{max} - \rho}{\tau_k} \tag{A2}$$

with $\rho_{max} = 350\;\mathrm{kg\,m^{-3}}$.

For dendritic snow we have

$$\frac{ds_p}{dt} = \frac{1 - s_p}{\tau_k} \tag{A3}$$

$$\frac{d\delta}{dt} = \frac{\delta}{2\tau_k} \tag{A4}$$

$$\frac{dd_{opt}}{dt} = \alpha\left[\frac{\delta}{2\tau_k}\,(s_p - 3) + (\delta - 1)\frac{1 - s_p}{\tau_k}\right] \tag{A5}$$

For non-dendritic snow

$$\frac{dd_{opt}}{dt} = -2\alpha s_p\frac{1 - s_p}{\tau_k} \tag{A6}$$

$$\frac{ds_p}{dt} = \frac{1 - s_p}{\tau_k} \tag{A7}$$





Redistribution of snow across model grid cell or tiles is not implemented at this stage; however we note that this issue is generally more relevant for local studies on small horizontal scales, rather than the grid cell scale of the climate-focused LM 4.1.

## Appendix B: Effect of snow compaction

Snow compaction is described following the approach used in the CROCUS model Vionnet et al. (2012). The mechanical settling of each snow layer $k = 1, \ldots n_L$ is computed as

$$\frac{d\Delta z_k}{\Delta z_k} = -\frac{\sigma_k}{\eta_k} dt \tag{B1}$$

With $\sigma_k$ the vertical stress due to the weight of overlying snow layers ($\sigma_j = \frac{1}{2} g \rho_k \Delta z_k + \sum_{j=1}^{k-1} g \rho_j \Delta z_j$, neglecting the effect of local slope, with $g = 9.806 \, \mathrm{ms^{-2}}$). The snow layer viscosity $\eta_k$ is parameterized as

$$\eta = f_1 f_2 \eta_0 \frac{\rho}{c_\eta} e^{a_\eta(T_F - T_i) + b_\eta \rho} \tag{B2}$$

with $\eta_0 = 7.62237 \times 10^6 \, \mathrm{kg \, s^{-1} \, m^{-1}}$, $a_\eta = 0.1 \mathrm{K^{-1}}, b_\eta = 0.023 \, \mathrm{m^3 \, kg^{-1}}$, and $c_\eta = 250 \, \mathrm{kg \, m^{-3}}$. The factors $f_1 = 1/(1 + 60 \, \theta_i)$ and $f_2 = \min(4, \exp(\min(g_1, 0.5g_s - g_2)/g_3))$ account respectively for the presence of liquid water and grain size effects on $\eta$. Here $g_1 = 0.4 \, \mathrm{mm}$, $g_2 = 0.2 \, \mathrm{mm}$, and $g_3 = 0.1 \, \mathrm{mm}$, $g_s = \alpha(4 - s_p)$ as in (Carmagnola et al., 2014), and $\theta_i = w_{l,i}/(\rho_w \Delta z_i)$ is the liquid water content of the snow layer.

## Appendix C: LM-CM BRDF snow albedo model

For the purposes of computing the surface albedo, in the case of little snow on the ground, fractional snow cover is computed based on snow depth as follows

$$f_{snow} = \frac{h_s}{h_s + h_{s,crit}} \tag{C1}$$

where $h_s$ is the total snowpack depth, and $h_{s,crit} = 0.0125 \, \mathrm{m}$ in a typical model configuration. Eq. (C1) is used in all model configurations compared in this paper. We remark that, as noted in (Dutra et al., 2010), this formulation for the effective snow fractional area to some extent accounts for the effect of snow density as it is based on snow depth rather than on SWE, so that the same mass of snow generally corresponds to a smaller fractional area in the spring snow melting phase.

The snow albedo model used in the current version of the model (GFDL LM 4.1 CM) is based on an empirical bidirectional reflectance distribution function (BRDF) by Schaaf et al. (2002). According to this model, black–sky albedo is expressed as

$$\begin{aligned}
\alpha_{bs}(\theta) = &f_{iso}\left(g_{0,iso} + g_{1,iso}\theta^2 + g_{1,iso}\theta^3\right) + \\
&f_{vol}\left(g_{0,vol} + g_{1,vol}\theta^2 + g_{1,vol}\theta^3\right) + \\
&f_{geo}\left(g_{0,geo} + g_{1,geo}\theta^2 + g_{1,geo}\theta^3\right)
\end{aligned} \tag{C2}$$



|            | iso  | vol       | geo       |
|------------|------|-----------|-----------|
| $g_0$      | 1.0  | -0.007574 | -1.284909 |
| $g_1$      | 0.0  | -0.070987 | -0.166314 |
| $g_2$      | 0.0  | 0.3077588 | 0.041840  |
| g          | 1.0  | 0.189184  | -1.377622 |
| f (cold)   | 0.92 | 0.06      | 0.0       |
| f (warm)   | 0.77 | 0.06      | 0.0       |

**Table C1.** Parameters of the BRDF model.

with $\theta$ the solar zenith angle. The white-sky albedo reads

$$\alpha_{ws} = f_{iso}g_{iso} + f_{vol}g_{vol} + f_{geo}g_{geo} \qquad (C3)$$

Given the ratio $r_{dif}$ of direct to total shortwave radiation received by the surface, the blue sky albedo is then obtained as $\alpha(\theta) = (1 - r_{dif})\alpha_{bs}(\theta) + r_{dif}\alpha_{ws}(\theta)$. Here the parameters $g_i$ are universal, given in (Schaaf et al., 2002) and reported in Table C1, while the $f_i$ depend on the specific surface properties. For snow, two set of parameters are used, corresponding to "cold snow", characterized by surface temperature below $10°$ C, and "warm snow" above freezing ($0°$ C). Within this temperature range the snow albedo is obtained by linear interpolation between that estimated for cold and warm conditions.

This formulation was introduced to mimic the effect of snow aging at different temperatures in the absence of information about snow microphysical properties.

*Author contributions.* All authors contributed to research design and manuscipt editing. E.Z. developed software, performed the simulations study, and drafted a first version of this manuscript. S.M. contributed to software development.

*Competing interests.* The authors declare no competing interests

*Acknowledgements.* The authors acknowledge funding from the NOAA Climate Program Office (CPO) grant number NA18OAR4320123 "3D-Land Energy and Moisture Exchanges: Harnessing High Resolution Terrestrial Information to Refine Atmosphere-to-Land interactions in Earth System Models". Funding support for this work was also provided through Interagency Agreement Number 80HQTR21T0015 from NASA Earth Science Division's High Mountain Asia Team program. The authors thank Dr. Justin Perket and Dr. Nicole Schlegel at NOAA GFDL for reviewing a first draft of the manuscript.



*Code and data availability.* The source code of GLASS v1.0 as well as the input data and model output are shared in a public repository:

`https://zenodo.org/records/10681526`



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
