# Peer review of "A Global land snow scheme (GLASS) v1.0 for the GFDL Earth System Model: Formulation and evaluation at instrumented sites"

_EGUsphere, 2024_

## Author Response (AR1)

**Response to reviewers**

We thank the reviewers for their helpful comments. We present our point by point response to the comments below. Referee comments are shown in black, and our response in blue. Line numbers in our response refer to the revised version of the manuscript unless otherwise stated.

In addition to the changes requested by the referees, we note that with respect to the first submission we have updated the online *zenodo* repository with model code, input data and results. The revised repository includes the entire land model necessary to replicate our analysis instead of just the snow model GLASS developed as part of this work.

https://zenodo.org/records/10901373

We are grateful to the reviewers for their constructive comments, which have significantly improved the manuscript. All comments have been carefully considered, and the manuscript modified accordingly. We have corrected multiple typos and clarified the text in multiple instances, particularly in section 3 which now we believe is more concise.

Enrico Zorzetto,
On behalf of all authors.

**Reviewer #1**

General comments

This paper presented a refined snow model (GLASS) and its implementation in the GFDL Earth system model. The GLASS model provides a detailed representation of the snowpack structure to track the evolution of snow grain properties in each snow layer. Testing cases were conducted using a reference dataset to evaluate the model performance. The results show that this new model improves the estimation of the soil temperature and have a better representation of SWE and daily snow albedo. The contribution of this work is worthy of publication, but it requires additional edits. Please see the detailed comments below.

We thank the reviewer for their comments. Please find below the response to each individual comment.

Detailed comments

1.      Pg 2 line 27: In the sentence "ranging from and watershed-scale ….", remove "and".

Revised as suggested.

2.     Pg 2 line 53-54: Please change the sentence as "It not only impacts the hydrological response but also interacts with the atmosphere through surface temperature and reflectivity."

Revised as suggested. The sentence now reads (line 56):

*"...as snow does not only impact the hydrological response, but also interacts with the atmosphere through surface temperature and reflectivity."*

3.     Section 3: Please enhance the notation used for the functions shown in this section. Some terms within the functions are not provided with full names and units when they first appear in this paper. The comments below only listed several places that need edits. So please review each function in this paper and make sure clear information is provided. This will help readers to understand and/or incorporate those functions into their model implementations.

We have now rewritten a large part of Section 3, improving the notation as suggested and rewriting some of the equations. We now make sure all variables and constants are appropriately defined when they are first used throughout the manuscript.

4.     Pg 7 line 179-185: Is the method for optimizing the vertical layers developed by the authors, or is it adapted from existing research? If it is developed by the authors, please provide the rationale behind the presented method. Specifically, why does it need to first define an optimal distribution of the snow layer and then compare it with the actual snow layers for adjustment? What's the benefits or reasons for it? Also, please explain why the optimal distribution of snow layers is defined as indicated in line 180-182. While the text describes how the optimal layers are defined, it lacks an explanation for why these specific numbers/parameters were chosen. Is there any scientific support for them?

We agree that the discussion of the layering methodology needed clarification. The layering scheme was indeed developed by the authors as part of the new snow model. We have now improved the description of the rationale behind the choice of the layering scheme and its parameters (at line 184):

*"The model optimizes the vertical layering of the snowpack by comparing the current layers with an optimal vertical distribution of snow layers defined for the current snow depth value. The optimal distribution of layers is designed with the objective of maintaining relatively thin layers close to the surface in order to better resolve heat diffusion and snowpack properties, and coarser layers at depth so as to limit the overall number of*

*layers. This is achieved by first specifying the optimal thickness of the top and the bottom layers. Below the first specified top layer, the optimal layers increase in thickness with a given constant ratio (set to 1.5 in the current model configuration), until they reach a specified maximum thickness (1 m in the default configuration used here). These parameter values were selected in order to restrain the number of layers, for computing efficiency, while allowing for relatively thin layers close to the surface, so as to better represent the vertical heterogeneity of snow properties and the temperature gradient close to the snow surface. There is no maximum number of snow layers set in the model."*

Furthermore, we have clarified how the merging and splitting of the layers is performed based on the optimal layer profile (at line 199):

*"The model loops through the existing snow layers, and for each layer compares the current value of $P_L$ with the corresponding metric evaluated after merging the current layer with the next. If after the merging of the layers the new value of the error metric is lower, the two layers are merged, unless the layers are not otherwise prohibited from merging because they have significantly different physical properties. Similarly, in a second loop GLASS attempts to split each snow layer in two by comparing the metric $P_L$ with the same metric relative to a new profile obtained by splitting a layer in two. Any time this comparison leads to a decrease in the metric $P_L$, the layer is split in two before examining the next."*

5.      Pg 11 function 13: please specify what "TF", "cl" and "λ" means.

We have revised the text and now define all parameters used in the manuscript, including those mentioned here. In particular, those mentioned here are the freezing temperature of water, the liquid water specific heat capacity, and snow thermal conductivity.

6.      Pg 11 function 14: please specify what "L0" means.

This variable represents the latent heat of sublimation of water. This equation is no longer present in the updated version of the manuscript, and we made sure all the variables used are defined in the revised manuscript where they first appear.

7.      Pg 11 function 15: please specify what "cs" means.

This quantity is the specific heat capacity of solid ice, which is now defined in the manuscript where it first appears.

8. Pg 11 function 16: please specify what "Lvap" means.

This is the latent heat of vaporization of water. We make sure we define all the variables used in the revised manuscript.

9. Pg 11 function 17: please specify what "Isn,s" means. Also, does this term need to be shown in the Figure 1?

This term is identically zero and has now been removed from the manuscript.

10. Pg 17 line 450: remove ")" before the first comma.

Revised as suggested.

11. Pg 17 line 451: change "byJordan" as "by Jordan".

Amended.

12. Pg 19 line 494: please spell the full name for "GSWP3" and add citation or URL link for this dataset (if available).

Revised as follows (now at line 522):

*"In this dataset, each site includes both in-situ meteorological forcing for the observation period (as reported in Table 2), and a locally-corrected Global Soil Wetness Project Phase 3 (GSWP3) forcing dataset (Menard et al., 2019) for the period 1980-2015."*

13. Pg 25 line 575: please reference Figure 10 at the end of the first sentence in this paragraph.

Revised as suggested (Now at line 620):

*"When examining error metrics in soil temperature, there are important differences between LM-CM and LM-GLASS (Figure10)."*

14. Pg 27 line 591: please change "Figure 10" as "Figure 11".

Revised as suggested (line 631):

*"As a representative example of the performance of the models in capturing temperature variations in the underlying soil, we show results for soil temperature at three depths for the Col de Porte site, for which observations are available (Figure 11)"*

15.    Pg 32 line 637: In the paper citation, remove "?" or add the correct citation.

Revised as suggested (675):

*"In such a case further research would be necessary to improve the current model with a focus on transitions between snow and no-snow areas, especially over complex terrain. Recent approaches used to model land surface heterogeneity could be useful for this purpose (Chaney et al., 2018; Zorzetto et al., 2023)."*

16.    Pg 33 line 660: add "than" in the sentence. ("temperature predictions were on average higher *than* those obtained…").

   Revised as suggested.

**Reviewer #2**

The manuscript: "A Global land snow scheme (GLASS) v1.0 for the GFDL Earth System Model: Formulation and evaluation at instrumented sites" by Zorzetto et al. describes a new, physics-based snow scheme to be used in land surface schemes. The scheme includes many physics-based snow processes like heat conduction, compaction and liquid water flow, which improve the performance in some key metrics like SWE, snow depth, etc. The model is validated on a high quality dataset of snow sites across the globe, in different climate regimes. Those testing sites include both canopy and non-canopy sites. The tests thus are extensive. I think the manuscript suits the journal well, and could be published. But there are quite a few issues that need attention. Mostly to improve clarity, but there are also a few occasions where the source of parameterizations is not really clear, as I'll point out below.

We thank the reviewer for their comments. Please find below the response to each individual comment.

Major points:

1) - Abstract: The abstract (as well as the conclusions for that matter) could benefit from some more quantitative measures of the improvements found. The only conclusion of what the improvements bring is the one line at the end: "We find that, when compared to previous version of GFDL snow model, GLASS improves predictions of seasonal snow water equivalent and soil temperature under the snowpack." I think this needs to be expanded on with some quantitative results. Similarly for the conclusions.

As suggested, we have added quantitative results to both abstract and conclusions.

In the abstract we now state the following (line 14):

*"We find that, when compared to the current GFDL snow model, GLASS improves predictions of seasonal snow water equivalent, primarily as a consequence of improved snow albedo. The simulated soil temperature under the snowpack also improves by about 1.5 K on average across the sites, while a negative bias of about 1 K in snow surface temperature is observed.*

In the conclusions section (at line 705):

*"The observed improvement in SWE predictions primarily originates from the updated snow optical properties, which in LM-GLASS explicitly depend on snow optical diameter and grain shape. We found that the BRDF parameterization used in LM-CM in general tends to underestimate albedo at the test sites (by about 0.05 for sites with no vegetation), and thus can lead to overestimating snow melt. The largest (and more consistent across sites) improvement in model skill was found for soil temperature, which was generally underestimated by about 1.5 degrees in LM-CM, suggesting an overestimation in heat conductance through the snowpack. In LM-GLASS, soil temperature predictions were on average higher than those obtained from LM-CM during the snow accumulation phase at all sites, and were generally closer to observations at sites where these were collected. The increased model complexity of LM-GLASS is due to a combination of refined vertical resolution and additional physical processes resolved in the model. These features lead to a computational cost increased by about 5.6% for the resulting land model (LM-GLASS) on average over the test sites."*

2) - Section 3.2: I have read the section multiple times, but I didn't manage to comprehend the differences between implicit and explicit melt. I would expect $dB/dT\_g$, the denominator in Eq. 6 to be 0 when the surface is in melting conditions. The sentence in L235/236: "The new temperature … heat diffusion process." I did not understand how this works. The terms

are not included in Eq. 12 for example. I think that this section needs some attention and rewriting, to improve clarity.

We have now updated several parts of section 3 with a focus on clarifying the snow energy balance, as well as the difference between the implicit and explicit melt schemes. We believe the confusion stems from the fact that we did not explicitly include in the original manuscript submission all the numerical details of the implicit numerical scheme.

With respect to the specific issue mentioned here: in Eq. (6) of our original submission, B is the net energy gained or lost at the surface as result of solving eq. (5). We note that $\partial B / \partial T_g < 0$ always: If the surface warms up, there always follows an increase in outgoing energy due to multiple flux terms (longwave radiation, sensible heat) which are monotonically increasing with $T_g$, so that the denominator of eq. (6) is never zero.

However, we decided to remove this equation from the manuscript. Since we did not include the entire numerical solution for the surface energy balance in the paper, we understand how adding this equation might have been confusing. Instead, we now explain the procedure we adopt and refer to relevant literature (Milly et al., 2014) where this specific procedure was developed. In the revised manuscript we now state (line 233):

*"However, the rate Mg of water melt or freeze at the surface of the snowpack (if present) or at the ground surface (if snow is absent) imposes a significant non-linearity in eq. (5), since this term is constrained by the amount of liquid or frozen water which can undergo phase change in the snowpack or in the upper soil layer. In this case, following the procedure by Milly et al. (2014), the single nonlinear eq. (5) is solved in order to obtain the change in temperature at the surface of the ground (or, of the snowpack if present, the temperature at the top of the snowpack), Tg, which in turn is used to obtain the tendencies of all other prognostic variables of the problem from the linearized system. The solution of eq. (5) uses the current liquid or solid mass available (either in the snowpack, if present, or in the uppermost soil level) to provide a constraint for the change of phase rate Mg. The new temperature Tg + ΔTg obtained by the solution of eq. (5) will then be propagated downward through the snowpack by solving implicitly the vertical heat diffusion process."*

With respect to improving the discussion on implicit and explicit melt: We have completely rewritten this subsection to make it clearer, and decided to move it after we discuss the snowpack energy balance so that it is now easier for us to explain to the reader what the issue is. The revised section reads (line 348):

*"In the following, we justify our choice of numerical method to solve the snow melt. The implicit numerical solution of heat conduction through the snowpack must occur in two steps: first the heat fluxes through the snow as well as their tendencies are calculated, and*

*only in a second step the temperature profile is updated layer by layer. It is therefore possible that, when updating the temperature profile, snow layers that were fully frozen become warmer than freezing, and similarly snow layers which contain water in liquid phase can experience temperatures below freezing. When this happens, the resulting change of phase is computed according to equations. (20) and (21). However, solving the phase change according to this procedure, which in the following we term "implicit melt", or IM, can be problematic in the presence of large time steps, which can produce appreciable temperature increments within the snowpack. In GLASS we use an alternative approach termed "explicit melt", or EM. When solving the nonlinear surface energy balance for the ground temperature balance, we simultaneously compute an estimate for the melt term (Mg) as discussed in Section 3.2. By adopting this approach, when we solve the vertical heat diffusion, the fluxes we obtain are already consistent with this tentative melt estimate. In the second heat diffusion step the vertical temperature profile is again updated as in the case of IM, and similarly additional change of phase can occur based on the phase and temperature of the existing snow layers. However, since the tentative melt Mg was already evaluated, these additional melt or freeze correction terms are generally smaller and therefore we expect the solution to be less dependent on the time step. Since Earth System Models generally run for relatively large time steps (30 minutes in our land model), in the following we investigate the behavior of the IM and EM approaches for different time steps, in order to test their suitability for our application."*

3) - Section 3.3: Also here, I think some improvements can be made. Eq. 14 is supposed to represent Eq. 5, as per the line preceding the equation. But then G in eq. 5 is not included, while the heat advection from liquid water is included in Eq. 14, which is not listed in Eq. 5. I assume that in Eq. 5, this term would take the role of energy advected in rain water. In fact Eq. 16 is basically Eq. 5, just with the term G on the left hand side, and including the energy from rain. Eq. 14 and Eq. 15 then seem superfluous. I think this all needs to be made more consistent, for clarity.

We revised the notation as suggested. The reason why we included all equations (14), (15), and (16) in the original submission was to show how the energy balance is treated close to the surface. However, based on this comment we have realized that having these equations here is unnecessary as effectively the surface boundary condition is given by eq. (5), i.e., the surface energy balance. We have now rewritten the subsection, which is more concise but still includes all the information necessary to describe how the energy balance is treated in GLASS. The manuscript was updated: We report in the following Section 3.4 from the revised manuscript [Box 1, lines 265-277]:

265 **3.4 Snowpack energy balance**

The vertical energy balance in the snowpack is expressed as

$$\frac{\partial H_{sn}}{\partial t} = \frac{\partial q_h}{\partial z} + S_z \tag{9}$$

with $H_{sn}$ being the energy content of the snow, $S_z$ the local source term due to shortwave radiation absorbed within the snowpack, and $q_h$ the vertical heat flux given by

270 $$q_h = -\lambda \frac{\partial T}{\partial z} + c_l q_l (T_l - T_F) \tag{10}$$

where $T_l$ the temperature of the vertical water flow of rate $q_l$, $T(z)$ is the local snow temperature ad depth $z$, $T_F$ is the freezing temperature of water, $\lambda$ is the snow thermal conductivity, and $c_l$ the specific heat of liquid water. The boundary condition at the top of the snow ($z=0$) is given by the net heat flux at the surface from eq. (5), and by the effective liquid precipitation rate $q_l|_{z=0} = f_l$ and its temperature $T_l|_{z=0} = T_{pr,l}$.

275 The bottom boundary condition (at $z=z_b$) between snow and soil reads

$$-\lambda \left(\frac{\partial T}{\partial z}\right)\bigg|_{z=z_b,\,soil} = \lambda \left(\frac{\partial T}{\partial z}\right)\bigg|_{z=z_b,\,snow} + c_l\,(T_l|_{z=z_b} - T_F)\,I_{sn,l} \tag{11}$$

and additionally $T|_{z=z_b,\,snow} = T|_{z=z_b,\,soil}$.

*Box 1: Lines 265-277 from the revised manuscript.*

4) - Section 3.7: It is written: "Therefore, this nonlinear interaction between heat diffusion and heat flux due to sublimating snow is accounted for by correcting the layer's temperature to ensure that energy is conserved when both processes are considered to occur simultaneously."

This sounds like the model is accounting for the energy associated with latent heat twice. If it is included in the surface energy balance (Eq. 5), then the only thing that sublimation would need to take care of, is the mass transfer that is associated with the latent heat exchange. There is no need to additionally modify the temperature of the layer. That was already taken care of via the latent heat flux in the upper boundary condition.

We understand that this may have been confusing, and now we have added a clearer and more extensive explanation of the rationale behind this correction. As explained in the manuscript, we solve sequentially 'dry' processes (vertical heat diffusion) and vertical mass fluxes. Since these are solved sequentially, the temperature must be adjusted to conserve energy, since in reality the change in mass and in temperature occurs simultaneously. There is therefore a higher order interaction terms due to change of mass and temperature (i.e., d(mass)*d(temperature) ) which is not accounted for in the linearized equation. This term is accounted for here. We now expanded on this in the manuscript. We report in the following an excerpt from Section 3.7 of the revised manuscript [lines 313-333]:

**3.7 Snow sublimation**

The rate of sublimation is computed by solving the nonlinear equation for the surface energy balance, eq. (5). In the case a snowpack is present, we assume that the entire water vapor flux comes from sublimation, even if liquid water is present in the snowpack. While this is a simplified assumption, we note that the amount of liquid water present in the snow layers is limited as we will discuss in Section 3.10. The sublimating ice is lost from the uppermost snow layer. In the model, the heat diffusion through the snowpack and sublimation are resolved separately in two consecutive steps. However, since in reality the two phenomena occur simultaneously, the change in heat content of the top snow layer associated with sublimating snow must account for the simultaneous change in the layer's temperature due to heat vertical diffusion. Therefore, this nonlinear interaction between heat diffusion and heat flux due to sublimating snow is accounted for in GLASS by correcting the layer's temperature to ensure that energy is conserved when both processes are considered to occur simultaneously. The change in

the snowpack energy content is given by two contributions due to mass lost from the uppermost layer, and by its change in temperature due to vertical heat diffusion. Due to the implicit numerical scheme used, evaporation and sublimation are linearized around the current temperature value, and their value depends on the surface temperature tendency. To ensure energy conservation, a temperature correction $\Delta^{*E}T_1$ must be applied to the snowpack uppermost layer, since evaporation is computed before the temperature is updated for the current model time step. This is given by

$$[c_l w_{l,1} + c_s(w_{s,1} + \Delta w_{s,1})](T_1 + \Delta T_1 + \Delta^{*E}T_1) - (c_l w_{l,1} + c_s w_{s,1})T_1 = (c_l w_{l,1} + c_s w_{s,1})\Delta T_1 + c_s \Delta w_{s,1}T_1 \tag{18}$$

where the left hand side is the change in energy content of the uppermost snow layer, and the two terms in the right hand side are the change in energy content of the layer due to temperature vertical diffusion, and that due to sublimation. In eq. (18), $\Delta T_1$ is the change in temperature in the top snow layer obtained by solving the vertical heat conduction equation, and $\Delta w_{s,1} = E_g \Delta t$ is the change in mass of the top snow layer due to sublimation. From eq. (18) we can solve for $\Delta^{*E}T_1$:

$$\Delta^{*E}T_1 = \frac{c_s \Delta w_{s,1}}{c_l w_{l,1} + c_s(w_{s,1} + \Delta w_{s,1})}\Delta T_1 \tag{19}$$

*Box 2: Lines 313-333 from the revised manuscript.*

We note that this is a nonlinear correction to the sublimation flux already evaluated based on the surface temperature at the beginning of the time step, and not a "double counting". Furthermore, we note that the energy balance is satisfied after this correction, and the actual sublimation flux in an implicit numerical scheme is dependent on the tendencies of the prognostic variables (ground temperature Tg and air specific humidity qc).

5) - Generally speaking, I think publications on model schemes should report on the mass and energy balance error they achieve. Thus, summing up all surface and soil energy fluxes, and comparing with the internal energy change of the snow cover. Similarly, summing up all mass fluxes at the upper and lower boundary, and comparing with the change in SWE in the model. This generally gives a good insight in the numerical quality of the model. I know it can be a lot of work to set up, but I think it's almost mandatory when discussing a numerical scheme.

We completely agree with the comment. LM4 and GLASS were developed for long climate simulations, and conserving mass and energy (as well as other model quantities as carbon and nitrogen) is paramount in this type of simulations. For this reason, mass and energy conservation are strictly enforced in our simulations. In case mass and energy balance is not conserved beyond a certain precision, the simulation stops. In the current experiments, we have used these strict conservation checks.

The maximum violation of energy conservation allowed at each model physics time step (30 minutes) is $10^{-7}$ *Kg/m2* for water, and $10^{-6}$ *J/m2* for energy. One can see that the maximum violations one could get in one year of simulation (17520 time steps) are very small compared to the typical mass and heat content of a snowpack. The reason why the model was developed with these strict mass and energy conservation criteria is that the model is developed for centuries-long climate simulations where violations of carbon, mass, and energy balance would lead to unacceptable drifts in the simulations.

We now mention this in the manuscript in the "Experimental setup" section at line 550:

*"As the model is designed for long climate simulations, it is important mass and energy are conserved with good accuracy throughout a simulation. Mass and energy conservation are strictly enforced in the model: In the current application, we run the model checking at each model physics time step (30 minutes) that conservation violations do not exceed $10^{-7}$Kg/m2 for water, and $10^{-6}$ J/m2 for energy during any time step".*

6) - Lastly, the scheme is only evaluated on the point scale, while its intended use is for the large scale (global) simulations. For the manuscript, this is fine, and a bit of discussion is provided here, but it feels too short. Is the additional computational and memory consumption from the GLASS scheme acceptable for large-scale simulations? For example, it is risky to have a variable number of snow layers without an upper bound, since that makes it hard to control memory usage in large models with many grid points. In the Conclusions, it is briefly mentioned that it is within computation constraints, but what constraints were set here for the model development?

We completely agree with this comment, and we now provide more details on this issue in the discussion. We now quantify the increase in computational cost of GLASS, and mention that global experiments will be needed to fully quantify the additional cost globally. (line 678):

*"Given the increased model complexity of LM-GLASS with respect to LM-CM, it is important to investigate the increased computational cost for the land model. The average increase in runtime for LM-GLASS compared to LM-CM is 7.4% (with a standard deviation of 7.1% across the test sites) when considering only the "fast" 30-minutes time step of the model, in which snow physics is resolved. When considering the change in runtime for the entire land model, we find that the increase in computational cost reduces to 5.6%. Furthermore, we note that for most of the sites here the model is run without vegetation, which is a computationally costly component of the land model. We expect therefore that in the presence of vegetation the relative cost of the LM-GLASS snow model will be lower. For example, when considering only the three BERMS vegetated sites, we found that introducing LM-GLASS increases the cost of the land model only by 3.6%. Note that these results are only indicative of locations dominated by seasonal snow. However, we note that as snow depth increases the thickness of the deeper snow layers also increases so that the number of snow layers in LM-GLASS increases relatively slowly with snow depth. We believe additional analysis would be beneficial to assess the performance of GLASS over different settings such as arctic regions and glaciers. "*

Minor comments

- Generally, a few language corrections are required. For example, articles seem to be often missing. For example L1: "Snowpack modulates". I think this should be "The snowpack modulates" or "Snowpacks modulate". Similar errors are present throughout. Also some wrongly placed parentheses for citations are present, like for example L306: "the parameterization by (Yen, 1981)" → "the parameterization by Yen (1981)"

Revised as suggested. We checked the entire manuscript for similar issues and corrected multiple instances.

- L49 "it has been recognized": a phrasing like that calls for appropriate citations. Please add some.

We agree and revised as follows (line 51):

*"Despite the need for this trade-off, it has been recognized that a number of physical processes impacting the evolution of the snowpack should be resolved in land surface models, as they can be relevant for large–scale hydrological studies and forcoupled climate simulations. These include the effect of thermal insulation (Cook et al., 2008; Lawrence and*

*Slater, 2010) and the effect on snow microphysics on albedo (Flanner and Zender, 2006; Vionnet et al., 2012; He et al., 2017)"*

- L151: "Design of snow layers" is unclear, please rephrase.

Rephrased as suggested (line 154):

*"A fine vertical discretization of the snowpack is key to resolve the vertical variation of snow physical properties which affect the overall snowpack mass and energy balance".*

- L194-200: This part is very hard to follow. Any chance to rephrase?

The paragraph was indeed difficult to understand. We rephrased the entire paragraph as follows (line 199):

*"The model loops through the existing snow layers, and for each layer compares the current value of $P_L$ with the corresponding metric evaluated after merging the current layer with the next. If after the merging of the layers the new value of the error metric is lower, the two layers are merged, unless the layers are not otherwise prohibited from merging because they have significantly different physical properties. Similarly, in a second loop GLASS attempts to split each snow layer in two by comparing the metric $P_L$ with the same metric relative to a new profile obtained by splitting the layer in two. Any time this comparison leads to a decrease in the metric $P_L$, the layer is split in two before examining the next."*

- Fig. 1 caption "solid contributions to runoff": not sure what is meant here, how can snow or ice runoff?

The term solid runoff was included in the paper for generality. This flux term is present in LM4.1 because there is an option in the model to cap the maximum amount of snow in a grid cell so that any amount of snow exceeding this threshold can be removed and become "solid" runoff. This strategy can be employed in coupled land-ocean-atmosphere simulations to prevent snow over glaciers from growing indefinitely. For this reason the term exists in the model, however this is not relevant for the current application, since this strategy is not used here. We therefore decided to remove mention of this term from the discussion of the model as this is not a feature of GLASS but one already existing in LM4.1 which does not affect our results and does not contribute to the discussion.

- L212: This is a bit confusing. In Section 3.7, it is argued that "we assume that the entire water vapor flux comes from sublimation". However, in Eq. 5, latent heat is associated with Lg, which is defined as the latent heat of evaporation. I would argue that if only sublimation is considered, this must be the latent heat of sublimation. Furthermore, one could argue that

in reality, evaporation takes precedence over sublimation, because of the smaller latent heat associated with it. Why is liquid water evaporation neglected?

We now clarify this point. In general, the land model allows for a combination of evaporation and sublimation to occur in the same time step. When snow is present, in GLASS we assume that sublimation only can occur from the surface. We included the liquid evaporation term for generality in eq. (21) as this equation is general and also applies for the uppermost soil layer when snow is not present on the ground. In the case of snow, this equation applies with $E_l = 0$. We have now revised the notation used in this section.. At line 314 we now state:

*"The rate of sublimation is computed by solving the nonlinear equation for the surface energy balance, eq. (5). In the case a snowpack is present, we assume that the entire water vapor flux comes from sublimation, even if liquid water is present in the snowpack. While this is a simplified assumption, we note that the amount of liquid water present in the snow layers is limited as we will discuss in Section 3.11."*

- L341-343: Please try to find alternative wording. I just don't comprehend the issue that is being described. Which temperature difference is this about?

We agree that this sentence was not clearly worded and has not been removed. The sentence was referring to the issue of implicit vs explicit melt calculation. This issue has now been clarified in the manuscript and it is discussed in our response to major comment number 4) , so we refer to that specific response:

We understand that this may have been confusing, and now we have added a clearer and more extensive explanation of the rationale behind this correction. As explained in the manuscript, we solve sequentially 'dry' processes (vertical heat diffusion) and vertical mass flux. Since these are solved sequentially, the temperature must be adjusted to conserve energy, since in reality the change in mass and in temperature occurs simultaneously. There is therefore a higher order interaction terms due to change of mass and temperature (i.e., d(mass)*d(temperature) ) which is not accounted for in the linearized equation. This term is accounted for here. We now expanded on this in the manuscript at line 311.

- L389: "look-up" table: how is this constructed, and where can it be found?

The look-up table used in the Flanner and Zender (2006) parameterization was obtained by directly contacting the author (Dr. Mark Flanner) and was redistributed with the GLASS snow code available in the online supplementary information.

In the manuscript at line 410:

*"...are parameters derived from a look-up table (here made available in the online assets) as functions of snow density, temperature, and temperature gradient. This parameterization was developed by Flanner and Zender (2006) using a physically based model describing the evolution of snow specific surface area due to dry aging."*

- Eq. 35-38: Where do these equations come from? And why are Eq. 37 and 38 the same? Generally, sphericity and dendricity change in opposite directions, so both change rates having the same sign is unusual.

These equations for dry snow metamorphism were developed by Brun et al. (1992) and are also used in more recent versions of the CROCUS model (Vionnet et al., 2012):

Indeed, in the case of weaker temperature gradients, the time evolution of dendricity and sphericity have opposite signs: As the snow ages, the dendricity decreases and sphericity increases, and snow grains tend towards a rounder shape (see equations. (35) and (36) in the manuscript, following the equation numbering of our original submission).

However, this is not the case in the presence of sharp vertical temperature gradients: In this case, the snow grains tend to become faceted crystals instead of spheres, so that the sphericity also decreases as the snow ages.

Therefore, in the GLASS implementation, these two behaviors are both possible and the appropriate one is selected based on the local value of the vertical temperature gradient.

We now have clarified the text and explain the behavior of these equations in the manuscript at line 440:

*"Note that in the case of weak temperature gradients, the time evolution of dendricity and sphericity have opposite signs: As the snow ages, the dendricity decreases while snow grains tend towards a rounder shape. However, this is not the case in the presence of sharp vertical temperature gradients: In this case, the snow grains tend to become faceted crystals instead of spheres, so that the sphericity also decreases with snow aging."*

- Eq. 39: how are the coefficients b0, b1, b2 determined? Are these related to snow properties, as described in L433-436? This is not so clear.

We now clarify that these parameters were obtained by He et al., (2018) by using a stochastic radiative transfer model (geometric-optics surface-wave approach). We have further clarified the discussion of the albedo model used in GLASS, correcting and defining some variables which were not defined in the original submission. We report here the revised section of the paper [lines 445-468 of the revised manuscript]:
* * *
445 ### 3.12 GLASS snow albedo model

In GLASS, in addition to the BRDF albedo model (see Appendix C) we employ the albedo parameterization proposed by He et al. (2018b) derived based on a stochastic radiative transfer model. In this formulation, snow albedo in the visible ($b = VIS$) and near infrared ($b = NIR$) bands is expressed as a function of snow grain shape and size as

$$\alpha_b = b_0\left(b, \delta_p, s_p\right) + b_1\left(b, \delta_p, s_p\right) R_n + b_2\left(b, \delta_p, s_p\right) R_n^2 - \Delta\alpha_b \tag{33}$$

450 where

$$R_n = \log_{10}\left(\frac{R_e \phi_b(\mu)}{R_0}\right) \tag{34}$$

with $R_e$ the snow grain effective radius defined as $R_e = 3V_s/(4A_s)$, where $V_s$ and $A_s$ are the snow grain volume, and the projection of its surface area average across all directions, respectively. $R_0 = 100\ \mu$ m is a reference snow grain effective radius. The grain radius can be related to the SSA as $R_{SSA} = 3/\rho_i/SSA$. For convex shapes $R_{SSA} = R_e$ while for Koch snowflakes

455 $R_{SSA} = 0.544 R_e$. The model parameters $b_0, b_1, b_2$ depend on the band and on the shape of the snow grains, and are tabulated in He et al. (2018b). The correction term $\Delta\alpha_b$ accounts for the effect of impurities deposited on old snow. While this phenomenon will be examined separately in future extensions of this study, here we use a simple correction similar to that used in Vionnet et al. (2012) for an alpine site conditions. We evaluate the decrease in visible albedo as $\Delta\alpha_{VIS} = \min\{0.2, 0.2\ age/60\}$. There is no correction in the near infrared band, so that $\Delta\alpha_{NIR} = 0$. In eq. (34), $\phi_b$ is a correction factor for direct light depending

460 on the cosine of the solar zenith angle $\mu$. While for diffuse light $\phi_b = 1$, in case of direct radiation the dependence of snow albedo on the direction of incident radiation is accounted for following Marshall (1989)

$$\phi_b(\mu) = \left(1 + a_{\theta,b}\Delta\mu\right)^2 \tag{35}$$

with $a_{\theta,b} = 0.781$ for visible band ($b = VIS$) and $a_{\theta,b} = 0.791$ for the near infrared band ($b = NIR$), and $\Delta\mu = \mu - \mu_D$, where $\mu = \cos\theta$ and $\mu_D = 0.65$ corresponds to $\theta = 49.5°$. Grain shape has been recognized to play an important role in determining

465 the optical properties of the snow medium (Robledano et al., 2023). He et al. (2018a) developed the parameterization defined by eq. (33) for different snow grain shapes, idealizing snow as a collection of either (a) spheres, (b) spheroids, (c) hexagons, or (d) Koch snowflakes. In GLASS, snow microphysical properties are represented through three variables: grain sphericity, dendricity and optical diameter. These three parameters are used to characterize the effect of snow grain shape on snow

*Box 3: lines 445-468 of the revised manuscript.*

- Eq. 40: R0 doesn't seem to be defined? How determined?

Revised as suggested. We now state that $R_0$ is the reference snow grain effective radius, set to 100 μm following He et al., (2018). At line 453: ".. $R_0$ = *100 μ m a reference snow grain effective radius.*"

- Section 3.13: I like the flow chart, but the readability of this section could be improved if the different elements in the flow chart are numbered or labeled (i), (ii), (iii) for example, such that the processes in the text can be assigned those labels or numbers to. This allows the text to be more precisely linked to the figure.

We revised as suggested. We now label all the steps in the flowchart (see the new version of Figure 2 reported here):

[Figure]

*Figure 2. A schematic representation of the main model steps in GLASS and their interface to other relevant physical processes in the GFDL LM4.1.*

When summarizing the model steps in the manuscript, we now explicitly refer to the panels as labeled in the figure as follows (lines 486-515 of the revised manuscript):

**3.13 Models steps summary**

Figure 2 provides a schematic representation of the computational steps performed to update the state of the snowpack at each model time step, summarized in panels A - G. Due to the nature of the implicit solution adopted for the energy and water balance, the heat diffusion through the snowpack must be solved in two separate steps. In a first snow model ("*step 1*", panel
490   B), the heat fluxes through the snowpack are computed starting from the lower boundary accounting for possible heat sources within the snowpack (e.g., due to shortwave radiation absorption). In this first step, an estimate of the ice available for melting is also computed. Then, the surface energy balance is performed, according to eq. (5) (panel D). Solving this equation yields the tendency for the surface temperature $\Delta T_g$ as well as the amount $M_g$ of melting ice or freezing water, depending on its sign. This information is then used in the second model step ("*step 2*", in panel E) of the snow energy and mass balance: The
495   temperature profile in the snow is first updated based on the upper boundary tendency $\Delta T_g$ and the vertical fluxes obtained in step 1 (panel E1). The mass of liquid and ice in the snowpack is then updated, based on the estimate of water changing phase ($M_g$) previously computed (panel E2). Note that after this step it is still possible that the solution of the heat equation yields above-freezing temperatures in some snow layers, or below-freezing temperatures in layers where liquid water is present, which are resolved with an additional change of phase. This implicit melt is then applied by evaluating the thermal equilibrium of
500   each snow layer (panel E3): In the case of layers with solid ice and temperature above freezing, a new equilibrium temperature is computed and the excess heat is used to melt part of the available ice. Conversely, in case of layers containing liquid water and below-freezing temperature, liquid water is frozen until thermal equilibrium is reached.

After performing the energy balance, the following steps are computed. Fresh snow is added to the snowpack in the presence of snowfall (E5). The model first tries to add the new snow to the existing uppermost layer. If the snowfall mass exceeds a
505   threshold, a variable number of layers is added to the top of the snowpack. We then perform the liquid balance in the snowpack (E6): Liquid precipitation is added to the top layer. The maximum liquid water capacity of the layer is computed as a fraction of the layer pore space, given by eq. (24). If this liquid water content is exceeded, the excess water flows vertically to the underlying layers. This step is followed by the sequential solution of the liquid water balance for all snow layers, down to the bottom of the snowpack. In panels E7 we perform in sequence snow compaction (described in Appendix B), wet and dry
510   snow metamorphism, and evaluate the effect of wind drift (see Appendix A). Finally, at each time step we also re-layer the snowpack in two steps (panels E4 and E8 in Figure 2), trying to merge or split existing layers based on their distance from the optimal layering structure for the given snow depth, as discussed in Section 3.1. After computing this second snow physics step, the new integral properties of the near-surface snow layer are computed. The near-surface layer is defined as the top three centimeters or the entire snowpack, whatever is smaller. These properties are then used to compute snow albedo (panel G) as
515   discussed in Section 3.12.

*Box 4: Lines 486-515 of the revised manuscript.*

- L477: "Finally, we re-layer …" I don't think this is shown in Fig. 2, but I think it should.

We agree. We have now added the snow relayering steps in Figure 2, and explained them in the manuscript, as discussed in our response to the previous comment.

- L505-508: I suggest this sentence, it's hard to follow now.

We rewrite the sentence as follows, explaining the purpose of the model spinup in our experimental setting. At line 541:

*"In order to perform a meaningful comparison between model and observations, we need to obtain a suitable initial condition for the state of the land model. This is done by performing a model spinup in which key land variables (e.g., vegetation if present, water and heat content in the soil) evolve driven by the atmospheric forcing observed at each site. We found that for all sites presented here in our model the soil is not frozen during the summer, and that the equilibration times characteristic for equilibrium is reached after less than 20 model years of model run."*

- L498-500: This is somewhat poorly phrased. At Col de Porte, the measurements are done at a constant height above the snow surface. Thus, the correction is done at the data level. For the other sites, the model doesn't correct the height of the measurements for the presence of a snow cover, when calculating heat fluxes. I think the phrasing should be more along the lines of this.

We agree, and revised the wording as recommended (line 533):

*"At the Col de Porte site, measurements are done at constant height above the snow surface. However, this is not the case for the other sites in the dataset, and the model does not correct for the varying height of the measurements above the surface when calculating turbulent fluxes at the snow surface."*

- L532-535: It's somewhat confusingly written. The sentence: "For example, for the swa site with some of the largest differences between the two models, the BRDF albedo scheme used in LM-CM leads to a significant underestimation of daily albedo (Fig. 5A)." Could be followed by "This underestimation is not present in the GLASS albedo scheme."

Revised as suggested by adding: *"This underestimation is not present in the GLASS albedo scheme."*

Then, the sentence: "For the three BERMS forested sites (ojp, obs, and oas), where the model simulates the effects of multi-layer canopy on radiative fluxes, the SWE predictions of

the two models are much closer (Fig. 5B). However, in this case modelled and observed albedo values differ significantly." Could be added "differ significantly in both GLASS and CM".

Revised as suggested:

*"However, in this case modelled and observed albedo values differ significantly in both GLASS and CM."*

- L549: I would avoid the term "snow amount", because it can cause confusion if it's about snow height or SWE.

Revised as suggested: we substitute "*snow amount*" with "*SWE*".

- Fig. 5 & 7 / Section 5.2: In the discussion on albedo, it's striking that when there is no snow, the model has a consistent bias compared to observations (looking at the summer months in Fig. 5). I know that it is discussed in the manuscript, but maybe it's better to restrict comparing albedo to the months with snow cover only in panel 7C. The GLASS model modifications seem to impact only snow physics, and not soil physics. Thus, the summer months are less relevant for the manuscript.

We agree that the difference is striking, and that the issue is outside the scope of our manuscript. We have updated the figure as follows, reporting albedo values only when the model predicts there is snow on the ground:

[Figure]

[Figure]

- L567-573: The discussion in L584-589 needs to be moved closer to L567-573. It could be mentioned more explicitly that an underestimation of surface temperature can also result from an underestimation in near surface density.

We have revised the manuscript as suggested: We moved this paragraph as suggested so that the discussion of the role played by thin snow layers directly follows the results on snow surface temperature. At line 617 we now state:

*"A potential reason for the colder snow surface values predicted by LM-GLASS is that the near-surface snow layers in LM-GLASS can be thinner than those in LM-CM, especially in the case of thick snowpacks. In this case, it is not surprising that thin surface layers with small heat capacity and increased insulating properties of the underlying snow layers would lead to a colder surface temperature. While this could be a limitation of LM-GLASS, it is also possible that cold temperatures at the surface originate from discrepancies between modelled and actual turbulent fluxes in the atmospheric surface layer. For all the snowpack variables, RMSE was also computed to complement bias, and is reported in Figure 9."*

- L637: seems to be missing a citation at the "?"

Revised as suggested:

*"In such a case further research would be necessary to improve the current model with a focus on transitions between snow and no-snow areas, especially over complex terrain. Recent approaches used to model land surface heterogeneity could be useful for this purpose (Chaney et al., 2018; Zorzetto et al., 2023)."*

**References**

Milly, P. C. D., and A. B. Shmakin. "Global modeling of land water and energy balances. Part I: The land dynamics (LaD) model." *Journal of Hydrometeorology* 3, no. 3 (2002): 283-299.

Shevliakova, E., S. Malyshev, I. Martinez-Cano, P. C. D. Milly, S. W. Pacala, P. Ginoux, K. A. Dunne et al. "The land component LM4. 1 of the GFDL Earth System Model ESM4. 1: Model description and characteristics of land surface climate and carbon cycling in the historical simulation." *Journal of Advances in Modeling Earth Systems* 16, no. 5 (2024): e2023MS003922.

Flanner, Mark G., and Charles S. Zender. "Linking snowpack microphysics and albedo evolution." *Journal of Geophysical Research: Atmospheres* 111, no. D12 (2006).

Brun, Eric, Paul David, Marcel Sudul, and Gilles Brunot. "A numerical model to simulate snow-cover stratigraphy for operational avalanche forecasting." *Journal of Glaciology* 38, no. 128 (1992): 13-22.

Vionnet, Vincent, E. Brun, S. Morin, A. Boone, S. Faroux, P. Le Moigne, E. Martin, and J-M. Willemet. "The detailed snowpack scheme Crocus and its implementation in SURFEX v7. 2." *Geoscientific model development* 5, no. 3 (2012): 773-791.

He, Cenlin, et al. "Impact of grain shape and multiple black carbon internal mixing on snow albedo: Parameterization and radiative effect analysis." *Journal of Geophysical Research: Atmospheres* 123.2 (2018): 1253-1268.

Chaney, Nathaniel W., Marjolein HJ Van Huijgevoort, Elena Shevliakova, Sergey Malyshev, Paul CD Milly, Paul PG Gauthier, and Benjamin N. Sulman. "Harnessing big data to rethink land heterogeneity in Earth system models." *Hydrology and Earth System Sciences* 22, no. 6 (2018): 3311-3330.

Zorzetto, Enrico, Sergey Malyshev, Nathaniel Chaney, David Paynter, Raymond Menzel, and Elena Shevliakova. "Effects of complex terrain on the shortwave radiative balance: a sub-grid-scale parameterization for the GFDL Earth System Model version 4.1." *Geoscientific Model Development* 16, no. 7 (2023): 1937-1960.